# MEAN-FIELD NEURAL DIFFERENTIAL EQUATIONS: A GAME-THEORETIC SEQUENCE PREDICTION

**Sungwoo Park**[*]   **Byungseung Kong**
Korea University
`{sungwoo_park, xncb135}@korea.ac.kr`

## ABSTRACT

We propose a novel class of neural differential equation models called *mean-field continuous sequence predictors* (MFPs) for efficiently generating continuous sequences with potentially infinite-order complexity. To address complex inductive biases in time-series data, we employ mean-field dynamics structured through carefully designed graphons. By reframing continuous sequence prediction as mean-field games, we utilize a fictitious play strategy integrated with gradient-descent techniques. This approach exploits the stochastic maximum principle to determine the Nash equilibrium of the system. Both empirical evidence and theoretical analysis highlight the unique advantages of our framework, where a collective of continuous predictors achieves highly accurate predictions and consistently outperforms benchmark prior works.

## 1 INTRODUCTION

Modeling spatiotemporal processes is central to understanding and predicting the behavior of complex systems that evolve across time and space. Recent work on neural differential equation models (Chen et al., 2019; Tzen & Raginsky, 2019) has shown that such architectures can effectively capture spatiotemporal dynamics in a wide range of applications, including generative modeling (Song et al., 2021), quantitative finance (Cohen et al., 2023), and physics-informed learning (Iakovlev et al., 2024). However, most of these approaches are formulated and evaluated under fixed, finitely sampled time grids, and therefore provide only limited theoretical insight into the following question about inherently continuous sequences: *How can we systematically model continuous-time sequences as the temporal discretization is refined and the effective number of events becomes very large?* In this work, we address this question by directly formulating the data dynamics in continuous time and then studying their behavior in the regime of increasingly fine temporal granularity. To obtain a tractable and theoretically grounded framework, we cast the prediction problem into the setting of *mean-field games* (Lasry & Lions, 2007), which gives rise to an infinite-dimensional predictive decision-making model that extends existing neural differential equation approaches (Tzen & Raginsky, 2019) to the analysis of continuous-time sequences.

The mean-field principle, a core philosophy in various scientific domains including neuroscience (Faugeras et al., 2009), statistical physics (Negele, 1982), and economics (Carmona, 2020; Cardaliaguet & Lehalle, 2018), serves as a powerful tool to model and analyze a large number of interacting agents, who behave in a manner that can be described as tragically rational within the decentralized coalition to satisfy *Nash equilibrium*. In this state of the mean-field regime, a continuum of infinitely many agents individually governs the dynamics of partially observed historical sequential data and collectively interacts with the others to make optimal group decisions for the prediction of future events. The foundational principle of this game-theoretic interpretation of the predictive system can be stated as follows: We reconstruct the continuous-time sequence prediction problem under the formal lens of mean-field games to gain powerful generalization capabilities in continuous sequence modeling. Stemming from the principle, we offer two main contributions:

- We extend conventional neural differential equation models by introducing mean-field principles, providing a new approach for modeling continuous sequences. This framework represents the stochastic spatiotemporal dynamics of an infinite continuum of agents and is rooted in hypotheses

---

[*]Corresponding author.

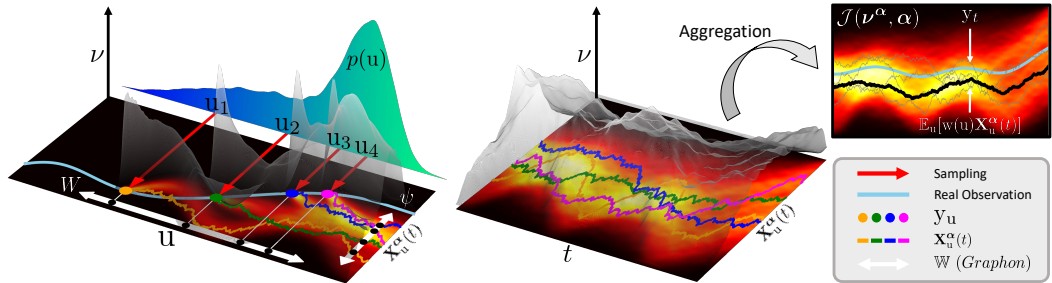

Figure 1: **(Left)**. The mean-field predictors are conditioned on a set of labeled past observations $\{\mathbf{u}_n\}_{n \leq N=4} \sim p(\mathbf{u})$. Each spatiotemporal dynamic is interconnected via the neural graphon $\mathbb{W}_{\boldsymbol{\alpha}}$, which leverages inductive biases tailored for continuous sequential data. **(Right)**. The collective decisions of a coalition of mean-field predictors are calibrated to approximate **(black trajectory)** the target future event interval.

from time series analysis (*e.g.*, seasonality). We demonstrate that our proposed method surpasses state-of-the-art benchmarks in continuous sequence prediction tasks including variants of state-space models.

• We propose a gradient-based mean-field FBSDE approach that provides feasible computational complexity for approximating Nash equilibria in mean-field games. Building on the concentration of empirical measures and the propagation of chaos property in the mean-field regime, our theoretical findings clarify the number of past observations on the generalization performance of the mean-field system. We demonstrate that the coalition produces increasingly accurate and reliable predictions.

**Problem Setup.** Given past observations $\{\mathbf{y}_\mathbf{u}\}$ in the interval $\mathbf{u} \in \mathbb{O} \subset \mathbb{T} = [0, T]$, the primary objective of the continuous sequence prediction task is to accurately forecast future events $\{\mathbf{y}_t\}$ within the interval $t \in \mathbb{T} \setminus \mathbb{O}$. The continuous sequences $\{\mathbf{y}_u, \mathbf{y}_t\} : [0, T] \to \mathbb{R}^d$ are continuously defined and share three notable properties: (1) *Irregularity*. The temporal granularity between spatio-temporal states in sequential data varies. (2) *Non-uniformity*. The cardinality of sequences exhibits stochastic and non-uniform behavior, fluctuating each time it is sampled from the dataset. (3) *Temporal scalability*. The sequential data spans multiple time scales, encompassing both short-term fluctuations and extended temporal ranges.

## 2 MEAN-FIELD CONTINUOUS SEQUENCE PREDICTORS

This section starts by introducing a stochastic differential equation model designed to depict infinite-order continuous signals, incorporating graphon structures for feature interactions.

**Definition 2.1.** *(Mean-field Graphon SDEs) For the Markovian feedback controls $\boldsymbol{\alpha} : \mathcal{T} \times \mathbb{R}^d \times \Theta \to \mathbb{R}^d$ (i.e., $\boldsymbol{\alpha} := \alpha(t, \mathbf{x}; \theta)$) and continuous labels $\mathbf{v} \sim p(\mathbf{u})$, we propose the $\mathbb{R}^d$-valued controlled stochastic differential equations called a mean-field graphon dynamics defined as follows:*

$$d\mathbf{X}_\mathbf{u}^{\boldsymbol{\alpha}}(t) = \langle \mathbb{W}_{\boldsymbol{\alpha}}[\nu_\mathbf{v}(t)](\mathbf{u}), \boldsymbol{\psi} \rangle (\mathbf{X}_\mathbf{u}^{\boldsymbol{\alpha}}(t), \boldsymbol{\alpha})dt + \boldsymbol{b}(t, \mathbf{X}_\mathbf{u}^{\boldsymbol{\alpha}}(t), \boldsymbol{\alpha})dt + \sigma_t dW_t^\mathbf{u}, \quad \mathbf{X}_\mathbf{u}^{\boldsymbol{\alpha}}(0) := \mathbf{y}_\mathbf{u}, \quad (1)$$

*where a probability measure $\boldsymbol{\nu} := \{\nu_\mathbf{v}(t)\}_{(\mathbf{v},t) \in \mathbb{O} \times \mathbb{T}}$ serves as a concise representation of the law of dynamics, and $\mathbf{y}_\mathbf{u} \sim p(\mathbf{u}, \mathbf{y})$ denotes a continuous representation of past observations.*

The mean-field dynamics presented in Definition 2.1 involves three terms on the right-hand side, with an emphasis on important notions *(A) mean-field predictors* and *(B) neural graphons*.

*(A) Mean-field Predictor.* The proposed dynamical system incorporates two types of continuity encoding: *locality* (*i.e.*, $t$) and *labeling* (*i.e.*, $\mathbf{u}$). The state variable $\mathbf{X}_\mathbf{u}^{\boldsymbol{\alpha}}(t)$, termed a *continuum of predictors* or *mean-field predictors* (**MFPs**), represent a continuous set of information flows, each labeled by $\mathbf{u} \sim p(\mathbf{u})$ and initialized from the past observation, $\mathbf{X}_\mathbf{u}^{\boldsymbol{\alpha}}(0) = \mathbf{y}_\mathbf{u} \sim p(\mathbf{u}, \mathbf{y})$. For instance, a continuum of predictors for the sequence of infinite i.i.d labels $\mathbf{u}_\infty := \{u_n \sim p(\mathbf{u}); n \leq N \to \infty\}$ in the mean-field regime $\mathbf{X}_{\mathbf{u}_\infty}^{\boldsymbol{\alpha}}(0)$ can be interpreted as being conditioned on the **past observational interval**, *i.e.*, the support of the label distribution $p(u)$, with their future causal outcomes, producing $\mathbf{X}_{\mathbf{u}_\infty}^{\boldsymbol{\alpha}}(t)$ at **future event interval** being obtained from the dynamics in Eq (1).

The suggested model effectively handles continuous signals by ensuring both input and output are processed continuously. Within this setting, the closed Markovian control process $\boldsymbol{\alpha}(\cdot; \theta) \in \mathbb{A}$,

parameterized by neural networks $\theta \in \Theta$, *neural agents*, controls the state dynamics $\mathbf{X}_{u_\infty}^{\alpha}(t)$. Fig 1 depicts illustrative examples of how the proposed mean-field predictors are conditioned (**left**), propagated (**mid**), and utilized to produce future prediction (**right**). The overarching goal is then to calibrate the trajectory of predictors by determining the optimal neural agent $\boldsymbol{\alpha}^*$ that closely approximates the target future interval, *e.g.*, $\mathbb{E}_t[\|\mathbb{E}_{u_\infty} \mathbf{X}_{u_\infty}^{\alpha^*}(t) - y_t\|_E^2] \approx 0$, where decision aggregation $w : \mathbb{O} \to [0, 1]$ captures the collective behavior of mean-field predictors. Section 3 will present a systematic algorithm to fulfill this objective.

*(B) Neural Graphon.* It is widely recognized in the literature that fundamental assumptions of inductive biases, such as *temporal decay*, *cycles*, and *seasonality* are vital for effective time series modeling. To incorporate these our mean-field system, we introduce a *neural graphon*, a graphon structure parameterized with neural networks, capturing the inherent heterogeneity among predictors.

---

**Definition 2.2.** *(Neural Graphon) A graphon is a set of symmetric integrable function $W : \mathbb{O}^2 \to \mathbb{R}$ equipped with $\mathbb{L}^2$-norm. For a probability measure $\mu$ defined on $\mathbb{O} \times \mathbb{R}^d$ with bounded second moment, we define a measure-valued function $\mathbb{W}_{\boldsymbol{\alpha}}[\mu](\cdot) : \mathbb{O} \to \mathcal{M}^a$ and a continuous symmetric function $\boldsymbol{\psi}_{\boldsymbol{\alpha}} := \boldsymbol{\psi}(y, x, \boldsymbol{\alpha}) := H_{\psi}(\boldsymbol{\alpha})\mathbf{Proj}(y - x)$ such that the first term in right-hand side of Eq (1) is defined as $\langle \mathbb{W}_{\boldsymbol{\alpha}}[\mu](u), \boldsymbol{\psi}_{\boldsymbol{\alpha}} \rangle(y, \boldsymbol{\alpha}) := \mathbb{E}_{v \sim p(v), x \sim \mu}[W_{\boldsymbol{\alpha}}(u, v)\boldsymbol{\psi}_{\boldsymbol{\alpha}}(y, x)] \in \mathbb{R}^d$.*

  $^a$Please refer to Section A.2 for the deatils.

---

For two tuples $(x, u) \sim \nu_u \otimes p(u)$ and $(y, v) \sim \nu_v \otimes p(v)$, a symmetric function $\psi$ estimates scaled relative dissimilarity between *spatial features* x and y. The neural agent, *i.e.*, $H_{\psi}(\boldsymbol{\alpha})$, then adjusts the importance of dissimilarity by rescaling projected vectors, *i.e.*, $\mathbf{Proj}$. Meanwhile, the neural graphon $W$ encodes a degree of interaction between temporal variables u and v. Among the various graphon designs available, we propose two structures informed by inductive biases specific to continuous time series. Note

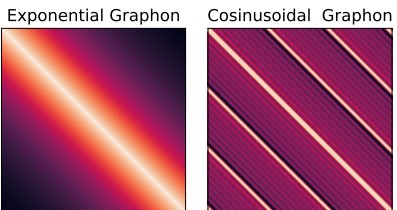

Exponential Graphon   Cosinusoidal Graphon

Figure 2: Visualization of Graphons.

that the key distinction from conventional methods is that our approach *directly models inductive biases in the data space* $\mathbb{R}^d$, rather than in latent feature spaces, facilitated by the graphon structure.

*Exponential Graphon.* In the first graphon structure, we incorporate *temporal decay* (Che et al., 2018) assumption on spatiotemporal variables, which suggests that the influence of the past event decreases exponentially as time deviations increase. An example depicted in Fig 2 illustrates an exponential graphon, highlighting that events occurring close in time often show strong interactions. Here, the function *i.e.*, $W_1 : \mathbb{A} \to \mathbb{R}^+$ determines the interaction magnitude. Subsequently, we introduce an exponential graph that diminishes the influence of events that are temporally distant; $W_{\boldsymbol{\alpha}}(u, v) := W_1(\boldsymbol{\alpha}) \exp(-T^{-1}\Delta_u)$ with $\Delta_u := |u - v|$.

*Cosinusoidal Graphon.* The second graphon is intended to highlight the continuous *cyclic* assumption (Oreshkin et al., 2020), which characterizes the periodic aspects of time-series data. To embody this assumption, we conduct an eigen-decomposition of the proposed graphon operator on $\mathbb{L}^2(\mathbb{O})$, employing sinusoidal eigen-functions (*i.e.*, $\{\psi_l\}$) and different frequency modes for the eigenvalues (*i.e.*, $\{\lambda_l\}$), as suggested by Gao & Caines (2019); $\mathbb{W} = \mathbf{Id} + \sum_{k,l \in \mathbb{Z}_+} \lambda_l \varphi_l$, where $\{\varphi_l\} \subset \{\mathbf{Id}, \sqrt{2}\cos 2\pi k, \sqrt{2}\sin 2\pi k\}$ and $\{\lambda_l\} \subset \{a_0, b_k/2\}$. We utilize neural networks to parameterize the graphon operator, substituting the Fourier coefficients $\{\mathbf{Id}, \lambda_l\}$ with equivalent neural networks, specifically $W_0, W_{1,l}, W_{2,l} : \mathbb{A} \to \mathbb{R}^+$. To illustrate different periodicities, we introduce $\mathfrak{f}(l) \in \{1/2, 1/4, 1/8\}_{l \leq L}$, which denotes a pre-determined series of frequencies. Consequently, we define a *cosinusoidal graphon* as follows:

$$W_{\boldsymbol{\alpha}}(u, v) = W_0(\boldsymbol{\alpha}) + \sum_{l \in \{1, \cdots, L\}} W_{1,l}(\boldsymbol{\alpha}) \cos(\cdot) + W_{2,l}(\boldsymbol{\alpha}) \sin(\cdot). \tag{2}$$

where $(\cdot) := 2\pi\mathfrak{f}(l)\Delta_u/|\mathbb{O}|$. Note that we limit the summation to finite modes (*i.e.*, $L$) for computational tractability. Fig 2 illustrates periodic interaction magnitudes for a predefined frequency setup. Further details on the implementation and their analysis can be found in the Appendix.

## 3 TRAINING MEAN-FIELD NEURAL NETWORKS

In the prior section, we introduced mean-field continuous sequence predictors based on SDEs that incorporate spatio-temporal interactions. Given that the mean-field system in Eq (2.1) is characterized as *controlled SDEs* with neural agents, we can define the objective function as a *stochastic control problem*. More precisely, our main aim is to reduce the cost functional $\mathcal{J}$ formulated for training neural agents for sequence prediction and to derive the *value function* $\mathcal{V}$:

---

**Definition 3.1.** *(Cost functional)[a] For the given neural graphon $\mathbb{W}_{\boldsymbol{\alpha}}$, and fixed set of admissible controls $\mathbb{A}$, the cost functional is defined as follows:*

$$\mathcal{V} := \inf_{\boldsymbol{\alpha} \in \mathbb{A}} \mathcal{J}(\boldsymbol{\nu}^{\boldsymbol{\alpha}}, \boldsymbol{\alpha}) = \inf_{\boldsymbol{\alpha} \in \mathbb{A}} \mathbb{E}_{\boldsymbol{\alpha}, \boldsymbol{\nu}, t} \left[ \|\mathbb{E}_{u \sim p(u)} \mathbf{X}_u^{\boldsymbol{\alpha}}(t) - y_t\|_E^2 + \mathbf{G}^{\boldsymbol{\alpha}} \right]. \tag{3}$$

*where $\mathbf{G}^{\boldsymbol{\alpha}} := \mathbf{G}(\mathbf{X}_u^{\boldsymbol{\alpha}}(T), \boldsymbol{\nu}^{\boldsymbol{\alpha}})$ represents the terminal cost at time $t = T$, and $w : \mathbb{O} \to [0, 1]$ is a decision aggregation function, satisfying $\int w(u)du = 1$.*

---
[a]Please refer to Section A.4 for the details on the definition.

---

To predict future values, mean-field predictors operate by generating a unified measure, specifically referred to as a temporal marginal of predictors $\mathbb{E}_{u \sim p(u)} \mathbf{X}_u^{\boldsymbol{\alpha}}(t)$. Here, the expectation accounts for the label u by amalgamating weighted outputs (*i.e.*, $w$) from a range of predictors $u \sim p(u) := w_{\#}[\mathbf{Unif}(\mathbb{O})](u)^1$, aiming to approximate the target continuous interval $\{y_t\}_{t \in \mathbb{T}}$. Figure 1 (**right**) presents a demonstration of the decision-making mechanism. In pursuit of producing target intervals, neural agents are conditioned to extract the *value function* $\mathcal{V}$, which describes the state where a continuum of players unites to collaboratively forecast the most favorable future occurrences.

The difficulty in addressing this problem arises because the neural agent affects the population of predictors $\boldsymbol{\nu}^{\boldsymbol{\alpha}}$, which, in turn, persistently alters the individual state variables through interactions facilitated by the neural graphon. In the literature, these types of problems are typically described as *(graphon) mean-field games* (Lasry & Lions, 2007; Caines & Huang, 2021). In this study, we propose a novel methodology to cast **the continuous sequence prediction problem through the lens of mean-field games**. Our main aim is subsequently to identify the most suitable *optimal control* $\boldsymbol{\alpha}^*$ that fosters the optimal response in the recursive interaction between $\mathcal{V}$ and $\boldsymbol{\nu}^{\boldsymbol{\alpha}}$. In our analysis, we explore the derivation of exact solutions $(\mathcal{V}, \boldsymbol{\nu}^{\boldsymbol{\alpha}^*})$ from optimal control profiles over time, by studying the subsequent system of PDEs within the mean-field regime:

---

**Definition 3.2.** *(Forward-Backward PDE System). For the obtained optimal neural agent $\boldsymbol{\alpha}^*$, exact solutions of the value function in Eq (3) can be obtained by solving the following system of PDEs:*

$$\partial_t \mathcal{V}(t, x) + \sigma_t^2/2 \Delta \mathcal{V}(t, x) + H(t, x, \partial_x \mathcal{V}, \nu_u(t), \boldsymbol{\alpha}^*) = 0,$$

$$\partial_t \nu_u^{\boldsymbol{\alpha}^*}(t) - \sigma_t^2/2 \Delta \nu_u^{\boldsymbol{\alpha}^*}(t) + \nabla \cdot \left[ \left( \boldsymbol{b}_W(x, \nu_u^{\boldsymbol{\alpha}^*}(t), \boldsymbol{\alpha}^*) + \boldsymbol{b}(t, x, \boldsymbol{\alpha}^*) \right) \nu_u^{\boldsymbol{\alpha}^*}(t) \right] = 0,$$

*where $\Delta$ and $\nabla \cdot$ denotes Laplacian and divergence operators, respectively. The stochastic Hamiltonian system $H$ is given by*

$$H(t, x_u, a, \nu, \alpha) := (\boldsymbol{b}_W(x_u, \nu, \alpha) + \boldsymbol{b}(t, x_u, \alpha)) \cdot a + \|\mathbb{E}_{u \sim p(u)} x_u - y_t\|^2,$$

*where $\boldsymbol{b}_W(x, \nu, \alpha) := \langle \mathbb{W}_{\boldsymbol{\alpha}}[\nu](u), \boldsymbol{\psi} \rangle(x, \alpha)$ is the graphon interaction term in Definition 2.2.*

---

A system of decoupled PDEs consists of the *Hamilton-Jacobi-Bellman* (**HJB**) equation and the *Fokker-Planck-Kolmogorov* (**FPK**) equation, which individually describes the propagation rules of the state variable and the value function over time. In mean-field equilibrium states, a set of PDEs are coupled as the law of the state variables $\mathbf{Law}(\mathbf{X}_u^{\boldsymbol{\alpha}}(t))$ matches $\nu_u(t)$ with marginal errors. This specific mathematical constraint can be formally expressed in the following definition:

---

[1]Here, $f_{\#}\mu$ denotes a push-forward probability measure of $\mu$ through function $f$.

Figure 3: Illustrative Algorithm of the Gradient System of FBSDEs.

**Definition 3.3.** *(Mean-field ϵ-Nash Equilibrium). We say that a continuous flow of measure $\nu_{\mathrm{u}}(\cdot)$ is an ε-equilibrium of (graphon) mean-field games if there exists a numerical constant $\epsilon > 0$ such that the following inequality holds:* $\sup_{\mathrm{u},t} \left[ \mathcal{W}_2^2(\nu_{\mathrm{u}}(t), \mathbf{Law}(\mathbf{X}_u^{\boldsymbol{\alpha}^*}(t))) \right] \precsim \mathbb{O}(\varepsilon)$, *where $\boldsymbol{\alpha}^* \in \mathbb{A}$ is an optimal control of the problem in Eq.* (3).

The mean-field equilibrium described in Definition 3.3 characterizes a scenario where a continuum of predictors is not incentivized to modify their policies $\boldsymbol{\alpha}^*$ to non-optimal counterpart $\boldsymbol{\beta}$, which induces marginal errors, *i.e.*, $\mathcal{J}(\boldsymbol{\nu}^{\boldsymbol{\beta}}, \boldsymbol{\beta}) \geq \mathcal{J}(\boldsymbol{\nu}^{\boldsymbol{\alpha}^*}, \boldsymbol{\alpha}^*)$. Here, the law of optimal mean-field predictors closely approximates the population $\nu_{\mathrm{u}}$ with marginal errors $\epsilon$. This coupling integrates the Hamilton-Jacobi-Bellman (HJB) and Fokker-Planck-Kolmogorov (FPK) equations, forming a *master equation*. Several numerical methods exist to approximate solutions to mean-field games including fixed-point iterations (Lauriere, 2021), and fictitious play (Min & Hu, 2021). However, these methodologies are typically constrained to linear quadratic dynamics, leading to computational intractability when confronting non-linearity (*e.g.*, neural networks). Additionally, numerical simulations for obtaining analytic solutions of this system of PDEs present significant challenges due to the curse of dimensionality in high-dimensional data spaces. The following section is dedicated to addressing these issues by leveraging the deep neural architecture.

### 3.1 GRADIENT SYSTEM OF NEURAL FBSDEs

Inspired by computational algorithms designed for fictitious play (Cardaliaguet & Hadikhanloo, 2017), we explore a *gradient descent*-based algorithm, which enables us to tackle solving MFGs by fusing deep neural architectures. To be more specific, we propose a gradient system of *forward-backward stochastic differential equations* (Bensoussan et al., 2013), which is adapted for reflecting the update of neural agents with respect to the gradient descent algorithm.

**Definition 3.4.** *(Gradient System of FBSDEs)[a]. For the fixed flow of measures $\nu_u(\cdot) : \mathbb{T} \to \mathcal{P}_2$ and the fixed label $\mathrm{u}$ at each stage $m$, we consider a family of processes $(\mathbf{X}_{\mathrm{u}}(t), \mathbf{Y}_{\mathrm{u}}(t), \mathbf{Z}_{\mathrm{u}}(t))$ that solves forward-backward stochastic differential equations with respect to the proposed graphon system in Eq* (1) *given as follows:*

$$d\mathbf{X}_{\mathrm{u}}^{m,\boldsymbol{\alpha}_m}(t) = \boldsymbol{b}_W^m dt + \boldsymbol{b}^m dt + \sigma_t dW_t^{\mathrm{u}}, \quad d\mathbf{Y}_{\mathrm{u}}^{m,\boldsymbol{\alpha}_m}(t) = -H^m dt - \mathbf{Z}_t^m \cdot dW_t^{\mathrm{u}},$$

$$\boldsymbol{\alpha}_{m+1} := \alpha\left(t, \mathbf{X}_{\mathrm{u}}^{m,\boldsymbol{\alpha}_m}; \theta^m - \mathbb{E}_{\mathbf{Y},t\leq T}\left[\gamma^m \nabla_\theta \mathbf{Y}_{\mathrm{u}}^{m,\boldsymbol{\alpha}_m}(t)\right]\right), \quad \nu_{\mathrm{u}} = \mathbf{Law}(\mathbf{X}_u^{m-1,\boldsymbol{\alpha}_{m-1}^*}),$$

*where $\gamma^m > 0$ is a learning rate of the gradient descent at $m$-th stage, and $\{\boldsymbol{\alpha}_m\}_m \subset \mathbb{A}$ is a set of admissible neural agents. Then, the triplet can be identified with $(\mathbf{Y}_{\mathrm{u}}(t), \mathbf{Y}_{\mathrm{u}}(T), \mathbf{Z}_{\mathrm{u}}(t)) = (\mathcal{J}, \mathbf{G}, (\partial_{\mathrm{x}}\mathcal{J})\sigma_t^{-1})$.*

---

[a]For the detailed description of the FBSDE system, please refer to the Definition A.2

The proposed gradient system can be decomposed by iterating a two-step procedure, *i.e.*, **(A)** and **(B)**, over a total of $M$ stages. Fig 3 illustrates the evolution of the mean-field predictors related to the updated parameters of neural agents $\boldsymbol{\alpha}_m$ across different stages $m$. The details of the two-step procedure are specified below.

**(A) Information Propagation.** Initially, the system disseminates the information to a continuum of players by utilizing the population information of the previous stage, where the forward and backward system of SDEs propagates information relating to the updated population, $\nu_{\mathrm{u}}$.

$$\nu_{\mathrm{u}} \longleftarrow \mathbf{Law}(\mathbf{X}_{\mathrm{u}}^{m-1,\boldsymbol{\alpha}_{m-1}^*}), \quad (\mathbf{X}_{\mathrm{u}}^m, \mathbf{Y}_{\mathrm{u}}^m) \sim \mathbf{Law}(\mathbf{X}_{\mathrm{u}}^m|\nu_{\mathrm{u}}) \otimes \mathbf{Law}(\mathbf{Y}_{\mathrm{u}}^m|\nu_{\mathrm{u}}). \quad (4)$$

Note that the backward dynamics is propagated in **reverse** direction starting from its terminal state $\mathbf{Y}_{\mathrm{u}}(T) = \mathbf{G}$ while the forward dynamics evolve in the **forward** direction from the initial state. This shows that the proposed FBSDEs parallel the PDE system described in Definition 3.2.

**(B) Update Control Profiles.** In the subsequent step, the neural agent $\boldsymbol{\alpha}^m$ is updated with respect to its parameter $\theta^m$ following the steepest direction of minimizing the values of backward dynamics $\mathbf{Y}_{\mathrm{u}}^m$. The backward dynamics, associated with the cost functional $\mathcal{J}$ as described in Proposition A.2, guide the updates of the parameters, allowing the mean-field predictors to gradually approximate the target interval. Since we have proposed an iterative algorithm to solve MFGs, the remaining part aims to provide convergence guarantees and highlight optimality conditions.

Proposition A.4 guarantees that the gradient system in Definition A.2 induces optimal neural agents $\boldsymbol{\alpha}^*$, which yield a feasible value function (*i.e.*, $\mathbf{Y}_{\mathrm{u}}^m(0) \xrightarrow{m\to\infty} \mathcal{V}$) where the optimality of the control is represented in the sense of the *Pontryagin stochastic maximum principle* (Yong & Zhou, 2012). Specifically, we have the following two results:

$$\lim_{m\to\infty} \boldsymbol{H}(\,\cdot\,,\boldsymbol{\alpha}_m) \approx \inf_{\boldsymbol{\alpha}\in\mathbb{A}} \boldsymbol{H}(\,\cdot\,,\boldsymbol{\alpha}), \; dt\otimes d\boldsymbol{\nu}, \; \mathcal{V} \approx \mathbf{Y}_{\mathrm{u}}^\infty(0) = \mathcal{J}(\boldsymbol{\nu}^{\boldsymbol{\alpha}\infty}, \boldsymbol{\alpha}_\infty). \tag{5}$$

The result illuminates that a pair $(\lim_{m\to\infty}\boldsymbol{\alpha}^m = \boldsymbol{\alpha}^*, \lim_{m\to\infty}\boldsymbol{\nu}^{\alpha_m} = \boldsymbol{\nu}^{\boldsymbol{\alpha}^*})$ solves both HJB and FPK equations in Def 3.2, assuring stochastic optimality. Having obtained the value function, the next step is to provide an explicit estimation of margin $\varepsilon$ in the convergence of mean-field equilibrium.

**Convergence to Mean-field Equilibrium.** To rigorously analyze the convergence to equilibrium in a distributional sense, we define two distinct operators, $\Phi$ and $\Psi : \mathcal{M} \to \mathcal{M}$, referred to as the *projector* and *updater*, respectively. Each operator corresponds to one of the two steps mentioned earlier, as illustrated in Fig. 3:

$$\Phi(\boldsymbol{\nu}^{\boldsymbol{\alpha}m}) := \{\mathbf{Law}(\mathbf{X}_{\mathrm{u}}^{\alpha_m}(t))\big|_{\boldsymbol{\nu}=\boldsymbol{\nu}^{\alpha_{m-1}^*}} \; ; \; t\in\mathbb{T}, \mathrm{u}\in\mathbb{O}\}.$$

$$\Psi(\boldsymbol{\nu}^{\boldsymbol{\alpha}m-1}) := \{\boldsymbol{\nu}^{\boldsymbol{\alpha}m} = \boldsymbol{\alpha}_{m-1}^*; \mathcal{V} = \mathcal{J}(\boldsymbol{\nu}^{\boldsymbol{\alpha}_{m-1}^*}, \boldsymbol{\alpha}_{m-1}^*)\}.$$

It can be easily verified that the composition of these operators at stage $m$ maps the previous state's population to the next stage *i.e.*, $\Phi\circ\Psi(\boldsymbol{\nu}^{m-1}) = \boldsymbol{\nu}^m$. Proposition 3.5 asserts that the population $\{\boldsymbol{\nu}^{\boldsymbol{\alpha}m}\}_{m\leq M}$ generated by the proposed algorithm begins to converge in the Wasserstein metric as the stages $m$ increase.

**Proposition 3.5.** *(informal) For arbitrary* $\mathrm{u} \sim p(\mathrm{u})$ *and* $t \in \mathbb{T}$*, the* $m$*-fold of composition* $\Phi\circ\Psi$ *induces convergent behavior of squared 2-Wasserstein distance:*

$$\mathcal{W}_2^2([\Phi\circ\Psi]^{\circ m}(\boldsymbol{\nu}^{\boldsymbol{\alpha}_1}), [\Phi\circ\Psi]^{\circ m}(\boldsymbol{\nu}^{\boldsymbol{\alpha}_0})) \precsim \sup_{t\in\mathbb{T}}\|\nabla_\theta\mathbf{Y}^m\|_E \cdot \mathbb{O}(\gamma^m, C) := \varepsilon_m \xrightarrow{m\to\infty} 0. \tag{6}$$

*where a numerical constant* $C$ *is dependent on* $M, b_0, C_1, H_{\boldsymbol{\psi}}, \mathbf{Lip_b}$ *and* $\mathfrak{m}_2, |\mathbb{O}|, e^{-|\mathbb{O}|}, \mathbf{Lip}_W,$ $\mathfrak{h}(\boldsymbol{\alpha}) = \|W_{\boldsymbol{\alpha}}\|_{\mathfrak{g}}$ *is a cut-norm of the proposed graphons (i.e., exponential, cosinusoidal)*

Proposition 3.5 reveals two theoretical implications regarding the convergence property. First, the proposed gradient system converges in a distributional sense, as the Wasserstein distance between the populations $([\Phi\circ\Psi]^{\circ m}(\boldsymbol{\nu}^{\boldsymbol{\alpha}_1}) = \boldsymbol{\nu}^{\boldsymbol{\alpha}m+1}$ and $([\Phi\circ\Psi]^{\circ m})(\boldsymbol{\nu}^{\boldsymbol{\alpha}_0}) = \boldsymbol{\nu}^{\boldsymbol{\alpha}m}$, governed by the gradient norm of the backward dynamics, is expected to decrease as $m$ increases. In other words, $\{\Phi\circ\Psi\}^{\circ m}$ is a Cauchy sequence in $\mathcal{M}$, ensuring the convergent behavior of the proposed training scheme. Second, the proposed gradient system ensures the convergence of the dynamics for the upper bounds $\epsilon_m$. It is important to note that the inequality in Eq (6) is an equivalent expression of the **mean-field Nash** $\varepsilon_m$**-equilibrium** described in Definition 3.3. In this context, the neural agent with greater capacity (*i.e.*, a smaller radius $\mathbf{r}_m$ of the metric balls in Eq (43)) further tightens the upper bound. In conclusion, the findings from Proposition 3.5 validate that the proposed gradient system efficiently utilizes neural networks to solve mean-field games in continuous sequence prediction.

## 4 SAMPLING MEAN-FIELD PREDICTORS

In this section, we propose the numerical algorithm for sampling the proposed mean-field predictors and provide a theoretical analysis of the sample complexity error and the asymptotic convergence of empirical estimation for mean-field predictors. Inspired by the Euler-Maruyama approach for Mckean-Vlasov type Reisinger & Stockinger (2022), we introduce an Euler-Maruyama method tailored for graphon-interacting particle systems to sample a series of mean-field predictors.

Algorithm 1 in Section A.5 details the computational procedure for sampling these mean-field predictors. We assume that $\boldsymbol{\alpha}^* := \alpha(\cdot; \theta^*)$ is nearly optimal, as defined by the $\varepsilon$-Nash equilibrium

derived from the mean-field gradient system associated with FBSDEs. The infinite-dimensional characteristics of the proposed system introduce inherent complexity challenges when mean-field predictors are sampled and applied to finite-dimensional, real-world datasets. When these sampled mean-field predictions aim to approximate their mean-field limits, an important question regarding sample complexity surfaces. To thoroughly investigate, we begin by defining the probabilistic description for both the sampled and model dynamics, as elaborated below:

$$\textbf{MFPs in Alg. 1}\ :\ \nu_t^N := \frac{1}{N}\sum_i^N \delta_{\mathbf{X}_i^n(t)},\ \ \textbf{MFPs with }\infty\textbf{-order}\ :\ \hat{\mu}_t := \mathbb{E}_{\mathrm{u}\sim p(\mathrm{u})}[\nu_{\mathrm{u}}(t)]. \quad (7)$$

where $\mathbf{X}_i^N(t) \sim \nu_t^N$ is sampled predictors, which can be obtained from implementing the Algorithm 1 and the weighted sum $\Lambda_t$ approximates true collective prediction made by mean-field predictors $\mathbb{E}_{\mathrm{u}}\mathbf{X}_{\mathrm{u}}^{\boldsymbol{\alpha}}(t) \sim \hat{\mu}_t$ in Eq (3).

**Proposition 4.1.** *(Sampling Complexity) For arbitrary* $\mathrm{u} \in \mathbb{O}$*, let* $\nu_t^N, \hat{\mu}_t$ *be probability measures defined in Eq (7). Then, there exist numerical constants* $\mathfrak{c}_4, \mathfrak{c}_7, \mathfrak{c}_8, \mathfrak{c}_9 > 0, w > 0$ *and* $\kappa > 0$ *such that the probability of squared* 2*-Wasserstein distance can be controlled as follows:*

$$\sup_{t\in\mathbb{T}} \mathbb{P}\left[\mathcal{W}_2^2(\nu_t^N, \hat{\mu}_t) \geq \epsilon\right] \leq \mathfrak{A}(\mathfrak{B} + \mathfrak{C} + \mathfrak{D}),\ \ \mathfrak{A} := \frac{2\mathfrak{c}_7^{3/2}}{\kappa}\exp(\mathfrak{c}_4 e^{\frac{1}{2}\mathfrak{c}_1 T})\left(e^{\kappa T} - 1\right) \vee \mathfrak{c}_9 \exp(-4\mathfrak{c}_8),$$

$$\mathfrak{B} := \frac{e^{-N\epsilon^2/4\mathfrak{c}}}{\epsilon^2}, \quad \mathfrak{C} := \frac{1}{72^4\epsilon\sqrt{N}}, \quad \mathfrak{D} := \frac{e^{-N\epsilon}}{N}\left(1 - \frac{128 w \mathfrak{h}(\boldsymbol{\alpha})}{N}\right)^{-d/8}.$$

The established inequality presents the relation between squared 2-Wasserstein distance and the number of samples $N$, the dimensionality of the data distribution $d$. The proof primarily draws on the findings presented in Bolley et al. (2007). It is important to note that the result also guarantees the proposed system benefits from the *propagation of chaos* (Chaintron & Diez, 2022), validating the asymptotic behavior of the sampled predictions generated by the mean-field predictors:

$$\sup_{t\in\mathbb{T}} \lim_{k\to\infty} \mathcal{W}_2^2\left(\textbf{Law}\left(\mathbf{X}_{i_1}^n, \cdots, \mathbf{X}_{i_k}^n\right), \otimes_{\{j=1,\cdots k\}}\nu_{j/n}(t)\right) \leq \Omega(N, k) \to 0$$

Proposition 4.1, A.5 and the inequality above align with the intuition that *as the number of predictors* $N$ *increases (and dimensionality* $d$*), the sampled dynamics converges more closely to the mean-field limit* $\hat{\mu}_t$ *and* $\nu_{\mathrm{u}}(t)$. In particular, the right-hand side of the inequality is influenced by two exponentially decaying terms, while the other term decreases at a polynomial rate, both showing short-tailed concentration with respect to the number of past observations. Overall, our theoretical findings highlight advantages of capitalizing on mean-field games: Rational individuals (*i.e.*, $\delta_{\mathbf{X}_i^n(t)}$) satisfying Nash equilibrium and conditioned on partial information (*i.e.*, $\mathbf{X}_i^n(0) = \mathrm{y}_{i/n}$) forms a coalition (*i.e.*, $\nu_t^N$), and the group decision is progressively refined to collaboratively solve the continuous sequence prediction problem. **As the coalition size increases, the resulting predictions become progressively more precise and reliable**. In Section 6, we conduct an ablation study to numerically verify these theoretical findings.

## 5 RELATED WORK

**Neural Differential Equation Models.** In recent years, neural differential equation models have gained attention for their ability to capture the dynamics of complex continuous sequences. Latent ODEs (Rubanova et al., 2019) extend standard RNNs to handle continuous signals by integrating neural ODEs with them. Kidger et al. (2020) introduced differential equation models based on controlled differential equations (Neural CDE) to address a key limitation of neural ODEs, where solutions depend solely on initial conditions and not on subsequent observations. Recently, Contiformer (Chen et al., 2024) was developed, combining neural ODEs and Transformers into a single framework. Another line of research integrates stochasticity by utilizing SDEs, particularly for time-series applications. For instance, Latent SDE (Li et al., 2020) encodes sequential data in the latent space using neural SDEs, while MaSDE (Park et al., 2023) employs a concept of stochastic differential games to analyze time series. Koshizuka & Sato (2023) proposed a regularized neural SDE based on the Lagrangian Schrödinger bridge, and Oh et al. (2024) introduced three stable types (classes) of neural SDEs: Langevin-type SDE, Linear Noise SDE, and Geometric SDE.

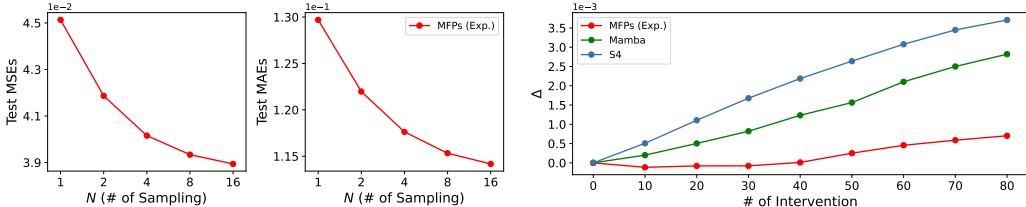

Figure 4: Ablation studies on EigenWorm dataset. (**Left**) Sensitivity analysis on the sample complexity. (**Right**) Robust analysis on post-intervention of non-informative signals.

| Methods | MIT Humanoid Robot | | MIMIC-II | | Beijing Air Quality | |
|---|---|---|---|---|---|---|
| | MSE | MAE | MSE | MAE | MSE | MAE |
| Neural Laplace | 8.11±0.25 | 17.03±0.33 | 7.76±0.04 | 18.70±0.08 | 3.21±0.12 | 11.45±0.23 |
| MaSDEs | 16.51±0.21 | 27.89±0.30 | 8.41±0.06 | 20.67±0.08 | 3.47±0.03 | 13.13±0.07 |
| CRU | 32.08±5.07 | 42.50±3.90 | 13.09±0.31 | 24.68±0.47 | 3.48±0.06 | 12.76±0.19 |
| Latent SDE | 6.01±0.14 | 15.94±0.14 | 8.04±0.02 | 19.63±0.06 | 3.29±0.03 | 11.99±0.07 |
| Neural LSDE | 6.80±0.14 | 16.51±0.08 | 7.93±0.05 | 19.09±0.07 | 3.74±0.04 | 11.98±0.15 |
| CONTIME | 6.88±0.29 | 16.60±0.25 | 12.29±0.14 | 25.26±0.12 | 5.15±0.17 | 15.86±0.27 |
| Contiformer | 5.94±0.23 | 15.29±0.26 | 7.90±0.12 | 19.05±0.18 | 3.25±0.10 | 11.48±0.16 |
| S4 | 5.59±0.16 | 13.98±0.19 | 13.24±0.01 | 24.79±0.30 | 3.95±0.15 | 12.35±0.17 |
| Mamba | 5.21±0.09 | 13.71±0.15 | 13.23±0.02 | 24.76±0.19 | 3.68±0.14 | 11.56±0.24 |
| Jamba | 5.13±0.13 | 13.32±0.20 | 9.71±0.09 | 21.37±0.06 | 4.03±0.10 | 13.04±0.20 |
| MFPs (Exp.) | **3.31±0.30** | **10.12±0.22** | 7.51±0.08 | **18.59±0.11** | 2.98±0.15 | **10.06±0.31** |
| MFPs (Cosin.) | 3.91±0.07 | 11.43±0.07 | **7.51±0.06** | 18.60±0.10 | **3.13±0.07** | 11.38±0.08 |

Table 1: Mean Squared Errors (MSEs) and Mean Absolute Errors (MAEs) in various continuous sequence prediction tasks. The top and second-top scores in each dataset are highlighted in bold and underlined, respectively. Each metric is scaled by $10^{-2}$.

**Mean-field Principles in Generative Models**. Recent works utilized the mean-field principle to model the infinitely many random particles in high-dimensional data space, where they interact with each other. In Liu et al. (2022), the Schrödinger bridge was incorporated to address mean-field games in order to approximate data distributions for large populations. Park et al. (2024) introduced the concept of propagation of chaos to generate data structures with exchangeable high cardinality such as 3D point clouds.

## 6 EXPERIMENTAL RESULTS

**Datasets.** In the experiments, we evaluate our results against benchmarks using the following datasets: *(i)* MIT Humanoid Robot (Li et al., 2024), *(ii)* MIMIC-II (Silva et al., 2012), *(iii)* Beijing Air Quality (Zhang et al., 2017), and *(iv)* EigenWorm (Bagnall et al., 2018). The MIT Humanoid Robot dataset contains the robot's state trajectories during various activities, such as running, jogging, and stepping in place, with 27 features describing these states. The MIMIC-II dataset, from the PhysioNet Challenge 2012, consists of time series data with 41 features representing the first 48 hours of a patient's ICU admission (e.g., $SaO_2$ and cholesterol levels). The Beijing Air Quality dataset contains time series data for six air pollution indicators, collected from 12 different locations in Beijing. The EigenWorm dataset comprises six features that characterize worm motion by projecting their shapes onto the six principal eigenworms, providing continuous sequences of 1500 length. To ensure consistent training, we apply either min-max and z-score normalization on each data instance.

**Benchmarks.** Given our focus on continuous sequence modeling, the benchmark baselines consist of various continuous models, including Neural Laplace (Holt et al., 2022), MaSDEs (Park et al., 2023), CRU (Schirmer et al., 2022), Latent SDE (Li et al., 2020), Neural LSDE (Oh et al., 2024), CONTIME (Jhin et al., 2024), and Contiformer (Chen et al., 2024). To further enhance the baselines, we also incorporate continuous state-space models, such as S4 (Gu et al., 2022), Mamba (Gu & Dao, 2024) and Jamba Lieber et al. (2024). Performance evaluation is carried out using mean squared error (MSE) and mean absolute error (MAE) metrics. Each model is executed five times, with the average scores and standard deviations reported.

**Quantitative Results.** The main table presents a performance comparison with benchmark methodologies across three datasets. The results show that the proposed MFPs consistently outperform other benchmarks by significant margins on all datasets. Notably, conventional neural differential equation models perform reasonably well on the MIMIC-II and BAQD datasets, where sequences are

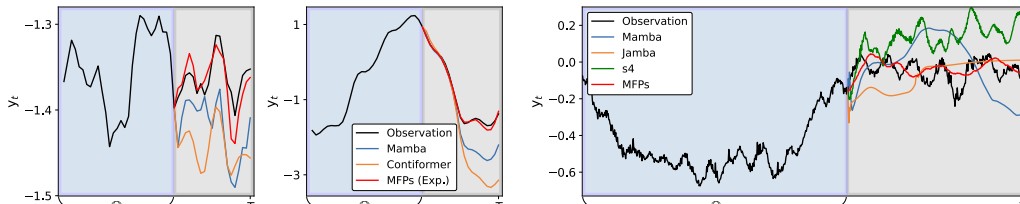

Figure 5: Qualitative results on Prediction **(Left)** MIT Humanoid Robot dataset **(Right)** EigenWorm datasets.

irregularly sampled with missing values. However, they exhibit a performance drop on the MIT Humanoid Robot dataset, likely due to their limitations in handling complex spatio-temporal dynamics. In contrast, state-space models excel on the MIT Humanoid Robot dataset but experience a decline in performance on the other two datasets, indicating their limitations in dealing with irregularly sampled sequences. Figure 4 (**right**) illustrates the qualitative prediction results on the MIT Humanoid Robot dataset. As shown, our MFPs deliver superior performance compared to the other models.

**Ablation Study I: Long-term Prediction.** The first study aims to demonstrate the efficacy of utilizing mean-field principles for accurate long-term predictions. We compare the MFPs against variants of state-space models, recognized for their capability in managing long-range dependencies Zhan et al. (2024) for long sequences. Table **??** along with Fig. 4 provide both quantitative and qualitative comparisons with benchmarks for long-term prediction tasks using the EigenWorm dataset. Evidently, our MFPs achieve significant advances over other benchmarks.

| Methods | EigenWorm | |
| | MSE | MAE |
| --- | --- | --- |
| S4 | 14.16±0.18 | 28.69±0.21 |
| Mamba | 15.79±1.03 | 29.80±1.06 |
| Jamba | 17.63±1.09 | 31.96±1.04 |
| MFPs | **12.52±0.16** | **26.61±0.29** |

Table 2: Long-term prediction on EigenWorm.

**Ablation Study II: Sample Complexity.** To empirically validate the theoretical results derived in Section 4, we conduct an ablation study examining how predictive accuracy scales with the number of sampled predictors. As depicted in Fig. 4 **(Left)**, our results align well with the concentration bounds formalized in Proposition 4.1, confirming that the empirical error diminishes as the coalition size of predictors increases. This behavior is consistent with the propagation-of-chaos property of mean-field systems, whereby larger predictor ensembles more faithfully approximate the infinite-agent limit distribution. In practice, this translates to monotonic improvements in both MSE and MAE as $N$ increases from 1 to 16, suggesting that additional predictors systematically enhance stability and reliability of forecasts. Nevertheless, these gains must be balanced against the computational overhead introduced during inference.

**Ablation Study III: Noise Robustness.** We perform a robustness study to assess the impact of non-informative noisy signal (*i.e.*, white noise) interventions in past observations. Specifically, we inject the Gaussian random noises with variance $\sigma_{\text{noise}} = 0.3$ to derive the distributional shift of test continuous-time sequences and corrupt the test data, $\hat{p}(\mathrm{u}, \mathrm{y}) = p(\mathrm{u}, \mathrm{y}) \circledast \mathcal{N}(\mathbf{0}_d, \sigma_{\text{noise}}\mathbf{I}_d)$, where $\circledast$ is a convolution operation. Fig 4 (right) shows a uniform performance degradation (*i.e.*, $\Delta$) with an increasing number of past observations corrupted by non-informative noisy signals. As can be seen, our MFPs exhibit robust performance against noise interventions, as Mamba experiences sharp declines in accuracy under high levels of noise. Our MFPs under the coalition, trained on the original clean sequence $p(\mathrm{u}, \mathrm{y})$, neutralizes the influence of individuals conditioned on noisy signals $\hat{p}(\mathrm{u}, \mathrm{y})$, thereby preserving the Nash equilibrium, resulting the robust generalization performance.

# 7 CONCLUSION

This paper introduces *mean-field continuous sequence predictors*, a novel class of neural SDE model for the efficient generation of continuous sequences, which can possess infinite-order complexity. We recast the time-series prediction problem as a mean-field game and adopt a fictitious play approach, integrated with a gradient-descent-based method, to leverage the stochastic maximum principle and identify the Nash equilibrium of the system. Both empirical and theoretical results reveal the distinctive features of our MFPs, where the coalition of a continuum of predictors generates accurate predictions and consistently surpasses benchmark performance.

## ACKNOWLEDGEMENT

This work was supported by ICT Creative Consilience Program through the Institute of Information & Communications Technology Planning & Evaluation(IITP) grant funded by the Korea government(MSIT) (IITP-2026-RS-2020-II201819)

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

# A  APPENDIX

## A.1  NOTATION TABLE

For convenience and to improve readability, Table 3 summarizes the main symbols used throughout the paper, together with brief descriptions of their roles in the mean-field formulation and learning algorithm.

Table 3: Summary of frequently used notation.

| Symbol | Meaning |
|---|---|
| $\mathbb{T} = [0, T]$ | Global time horizon of a sequence instance |
| $\mathcal{O} \subset \mathbb{T}$ | Past observation interval used as input |
| $u$ | Time label for past observations, $u \in \mathcal{O}$ |
| $t$ | Prediction time index, $t \in \mathbb{T}$ |
| $p(u)$ | Distribution of observation times on $\mathcal{O}$ |
| $p(u, y)$ | Joint distribution of times and observed values |
| $y_u, y_t$ | Observed value at time $u$ or $t$ |
| $d$ | Data dimension of each observation |
| $X_u^\alpha(t)$ | Mean field predictor at $(u, t)$ under control $\alpha$ |
| $\nu_u(t)$ | Law of $X_u^\alpha(t)$ at time $t$ for label $u$ |
| $\nu^\alpha$ | Flow of laws $\{\nu_u(t)\}_{u,t}$ induced by control $\alpha$ |
| $\hat{\mu}_t$ | Mean field limit obtained by averaging over $u$ at time $t$ |
| $N$ | Number of sampled predictors used in the finite system |
| $\nu_t^N$ | Empirical law of $N$ predictors at time $t$ |
| $\mathcal{P}_2(\mathbb{R}^d)$ | Probability measures with finite second moment on $\mathbb{R}^d$ |
| $W_\alpha(u, v)$ | Neural graphon weight between labels $u$ and $v$ |
| $\psi_\alpha(y, x)$ | Feature interaction function between states $y$ and $x$ |
| $b(t, x, \alpha)$ | Drift of the controlled SDE at $(t, x)$ |
| $\sigma_t$ | Diffusion scale of the SDE at time $t$ |
| $W_t$ | Brownian motion driving the stochastic dynamics |
| $\alpha(t, x; \theta)$ | Feedback control policy parameterized by $\theta$ |
| $\theta$ | Trainable neural network parameters |
| $\mathcal{A}$ | Set of admissible control policies |
| $\Lambda_t$ | Aggregated prediction obtained by averaging over labels $u$ |
| $J(\nu^\alpha, \alpha)$ | Cost functional of the mean field control problem |
| $V(t, x)$ | Value function associated with the control problem |
| $H(t, x, a, \nu, \alpha)$ | Stochastic Hamiltonian in the Pontryagin principle |
| $W_2(\cdot, \cdot)$ | 2-Wasserstein distance on $\mathcal{P}_2(\mathbb{R}^d)$ |
| $W_{t,\mathcal{M}}(\cdot, \cdot)$ | Generalized Wasserstein distance between flows of laws |
| $\Phi$ | Projector operator that updates the state law forward in time |
| $\Psi$ | Updater operator that refines the control via FBSDE |
| $m$ | Fictitious play or gradient descent iteration index |

## A.2 MATHEMATICAL BACKGROUNDS AND DEFINITIONS

This section includes brief summary of the mathematical backgrounds, omitted notations and definitions in the manuscript. Throughout, **bold-face** notation will be employed without loss of generality to omit subscript and superscript indices of mathematical entities where suitable for the sake of simplicity.

**Generalized Wasserstein Distance.** Recall the definition of space for probability measures that consist of generic path measures with finite second moments,

$$\hat{\mathcal{M}} := \{\boldsymbol{\nu} = (\nu_u : u \in \mathbb{O}) \in [C([0,T], \mathbb{R}^d)]^{\mathbb{O}}; u \mapsto \nu_{\mathrm{u}} \in \mathcal{P}(C([0,T], \mathbb{R}^d) \text{ is measurable}\},$$

$$\tilde{\mathcal{M}} := \{\boldsymbol{\nu}; \sup_{\mathrm{u} \in \mathbb{O}} \int \|\mathbf{X}_{\mathrm{u}}(\cdot)\|^2 d\nu_u(\mathbf{X}_{\mathrm{u}}(\cdot)) < \infty\}.$$

For the arbitrary elements $\boldsymbol{\mu}, \boldsymbol{\nu} \in \mathcal{M} := \hat{\mathcal{M}} \cap \tilde{\mathcal{M}}$, let us consider $\mathcal{M}$ equipped with the generalized 2-Wasserstein metric as

$$\mathcal{W}_{t,\mathcal{M}}(\boldsymbol{\mu}, \boldsymbol{\nu}) := \sup_{\mathrm{u} \in \mathbb{O}} \left[ \inf_{\Pi} \mathbb{E} \left( \sup_{s \leq t} \|\mathbf{X}_u(s) - \mathbf{Y}_{\mathrm{u}}(s)\|^2 \right) \right]^{1/2}, \quad \begin{cases} \mathbf{Law}(\mathbf{X}_{\mathrm{u}}) = P_u^{-1} \circ \boldsymbol{\mu}, \\ \mathbf{Law}(\mathbf{Y}_{\mathrm{u}}) = P_u^{-1} \circ \boldsymbol{\nu}, \end{cases} \quad (8)$$

where $\Pi$ is a coupling between two probability measures and $P_u$ denotes a canonical projection onto the interval $\mathbb{O}$. Followed by the Kantorovich-Rubinstein duality, definition in Eq (8) can be further modified as

$$L\mathcal{W}_{t,\mathcal{M}}(\boldsymbol{\mu}, \boldsymbol{\nu}) \geq \sup_{\mathrm{u} \in \mathbb{O}} \sup_{f \in \mathbf{Lip}(L)} \left| \int_{\mathbb{R}^d} f d(\mu_{\mathrm{u},t} - \nu_{\mathrm{u},t}) \right|, \quad \boldsymbol{\mu}, \boldsymbol{\nu} \in \mathcal{M}. \quad (9)$$

Note that the inner supremum is taken over a family of $L$-Lipschitz real-valued continuous functions.

**Cut Norm of Graphon**. The cut-norm measures the *discrepancy* between two graphons over all possible cuts of the square of $\mathbb{O}$. Formally, for a graphon $W : \mathbb{O} \times \mathbb{O} \to \mathbb{R}^+$, the cut-norm is defined as:

$$\|W\|_{\mathfrak{g}} := \sup_{A,B \subset \mathbb{O}} \left| \int_{A \times B} W(\mathrm{u}, \mathrm{v}) d\mathrm{u} d\mathrm{v} \right|, \quad (10)$$

where the supremum is taken over all measurable subsets $A$ and $B$. The definition illustrates that the cut-norm quantifies the maximum deviation of $W$ from zero over any rectangle $\mathbb{O}^2$. Given the definition, one defines the metric called *cut distance*:

$$d_{\mathfrak{g}}^q(W_1, W_2) = \|W_1 - W_2\|_{\mathfrak{g}}^q \quad (11)$$

The cut distance measures how close two graphons are after optimally aligning their domains. If the cut distance between two graphons $W_1$ and $W_2$ is small, the graphs they represent are structurally similar.

**(Exponential AM-GM Inequality).** For the arbitrary random variables $X, Y$ and positive constants $a, b > 0$, the expectation can be decomposed as follows:

$$\mathbb{E}[\exp(aX^2 + bY^2)] \leq \left(2\mathbb{E}[\exp(2X^2)]\right)^{1/2} \left(2\mathbb{E}[\exp(2X^2)]\right)^{1/2}. \quad (12)$$

**(Arithmetic AM-GM Inequality).** For arbitrary positive constants $x, y, w > 0$, we have

$$xy \leq \omega x + \frac{1}{4\omega} y. \quad (13)$$

### A.3 ASSUMPTION

Without additional information, we make the following assumptions in this paper.

1. (**H1**). There exists a finite collection of intervals $\{O_k; k \in \{1, \cdots, N\}\}$ for arbitrary $N \in \mathbb{N}^+$ such that $\cup_k^N O_k = \mathbb{O}$. Then we assume the following:

   - For each $k$, the initial datum of the graphon system is set with the data distribution $\nu_{\mathrm{u}}$: $O_k \ni \mathrm{u} \mapsto \mu_{\mathrm{u}}(0) := \nu_{\mathrm{u}} \in \mathcal{P}_2$, where the mapping assigns to independent measures.

2. (**H2**). For each $k$ and $O_k \ni \mathrm{u}$, there exists a constant $C_1$ such that we have probability $\nu_{u,s}[\sup_{x \in \mathbb{R}^d \setminus \mathbf{Y}(\omega)} \|\mathrm{x} - \mathbf{Y}\|^{-p} \leq C_1]$ almost surely for all $p \in \mathbb{N}^+$, and the second moment (*i.e.*, $\mathfrak{m}_2$) of $\nu_{u,s}$ is bounded.

3. (**H3**). The Lipschitz constants of the functions in modeling of graphons $W_{(\cdot)} : \mathbb{L}_2(\nu_{\mathrm{u}}(t)) \supset \mathbb{A} \to \mathbb{R}^+$ are bounded above. The parameterized Markovian feedback controls are Lipschitz in parameters:

$$|W_{(\cdot)}(\boldsymbol{\alpha}) - W_{(\cdot)}(\boldsymbol{\beta})| \leq \mathbf{Lip}_W \|\boldsymbol{\alpha} - \boldsymbol{\beta}\|_{\nu_{\mathrm{u}}(t)}, \tag{14}$$

$$\{\|\boldsymbol{\alpha} - \boldsymbol{\beta}\|_{\nu_{\mathrm{u}}(t)}, \|\alpha(t, \mathrm{x}, \theta_{\boldsymbol{\alpha}}) - \alpha(t, \mathrm{x}, \theta_{\boldsymbol{\beta}})\|\} \leq \mathbf{Lip}_\theta \|\theta_{\boldsymbol{\alpha}} - \theta_{\boldsymbol{\beta}}\|_E \tag{15}$$

The drift function is Lipschitz continuous and dissipative, ensuring that the constant $\mathfrak{c}_1$ is well-defined.

$$\|(\boldsymbol{b}, \boldsymbol{b}_W)(t, \mathrm{x}, \boldsymbol{\alpha}) - (\boldsymbol{b}, \boldsymbol{b}_W)(t, \mathrm{y}, \boldsymbol{\beta})\| \leq \mathbf{Lip}_{\boldsymbol{b}}(\|\mathrm{x} - \mathrm{y}\|_E + \|\boldsymbol{\alpha} - \boldsymbol{\beta}\|_{\nu_{\mathrm{u}}(t)}). \tag{16}$$

$$\mathfrak{c}_1 := \inf_{\mathrm{x,y}} -(\mathrm{x} - \mathrm{y}) \cdot [(\boldsymbol{b}, \boldsymbol{b}_W)(\mathrm{x}) - (\boldsymbol{b}, \boldsymbol{b}_W)(\mathrm{y})] / \|\mathrm{x} - \mathrm{y}\|_E^2 \tag{17}$$

4. (**H4**). The maximal rank of embedding of neural agents in $\mathbb{A}$ is $d'$.

$$\mathbb{T} \times \mathbb{R}^d \times \Theta \mapsto \boldsymbol{\alpha} \in \mathbb{A} \hookrightarrow \mathbb{L}_2(\boldsymbol{\nu}). \tag{18}$$

### A.4 PROOFS

#### A.4.1 STOCHASTIC OPTIMAL CONTROL, MEAN-FIELD FBSDEs

Before presenting the main proofs, this section offers a detailed analysis of how the proposed mean-field games can be formulated.

**Weak Formulation of Mean-field Games.** We start by explicating on the rigorous definition of forward mean-field dynamic in Eq. (1) cost functional in Eq. (3) and gradient system of FBSDEs in Propsoition A.2, followed by a brief summary of how forward-backward SDEs are formulated in the context of stochastic optimal control problems. To this end, let us first define the primitive process $\bar{\mathbf{X}}_t$, which solves the following SDE for a fixed label $\mathrm{u}$:

$$d\bar{\mathbf{X}}_{\mathrm{u}}(t) = \sigma_t dB_t^{\mathrm{u}}, \quad \bar{\mathbf{X}}_0(t) = \mathrm{y}_t. \tag{19}$$

where $B_t^{\mathrm{u}}$ is a Brownian motion under probability measure $\mathbb{P}$. Given the square of volatility term $\sigma_t^2$ is bounded below some constant, we introduce the probability measure $\mathbb{P}^{\boldsymbol{\mu}, \boldsymbol{\alpha}}$, which can be derived by the following Radon-Nikodym derivative:

$$\frac{d\mathbb{P}^{\boldsymbol{\mu}, \boldsymbol{\alpha}}}{d\mathbb{P}} = \mathcal{E}\left(\int_0^{(\cdot)} \sigma_t^{-1} \left(\boldsymbol{b}_W(\bar{\mathbf{X}}_{\mathrm{u}}(t), \boldsymbol{\nu}, \boldsymbol{\alpha}) + \boldsymbol{b}(t, \bar{\mathbf{X}}_{\mathrm{u}}(t), \boldsymbol{\alpha})\right) \cdot dB_t^{\mathrm{u}}\right)\Bigg|_{t=T}. \tag{20}$$

where $\mathcal{E}$ denotes a Doléans-Dade exponential of a martingale. Applying Girsanov's theorem, we have the Brownian motion $W^{\boldsymbol{\mu}, \boldsymbol{\alpha}}$ under the probability measure $\mathbb{P}^{\boldsymbol{\mu}, \boldsymbol{\alpha}}$:

$$W_t^{\mathrm{u}, \boldsymbol{\mu}, \boldsymbol{\alpha}} = B_t^{\mathrm{u}} - \int_{\mathbb{T}} \sigma_s^{-1} \left(\boldsymbol{b}_W(\bar{\mathbf{X}}_{\mathrm{u}}(s), \boldsymbol{\nu}, \boldsymbol{\alpha}) + \boldsymbol{b}(s, \bar{\mathbf{X}}_{\mathrm{u}}(s), \boldsymbol{\alpha})\right) ds. \tag{21}$$

Then, the primitive process can be rewritten as follows almost surely $\mathbb{P}^{\boldsymbol{\mu}, \boldsymbol{\alpha}}$,

$$d\bar{\mathbf{X}}_{\mathrm{u}}(t) = \left(\boldsymbol{b}_W(\bar{\mathbf{X}}_{\mathrm{u}}(t), \boldsymbol{\nu}, \boldsymbol{\alpha}) + \boldsymbol{b}(t, \bar{\mathbf{X}}_{\mathrm{u}}(t), \boldsymbol{\alpha})\right) dt + \sigma_t dW_t^{\mathrm{u}, \boldsymbol{\mu}, \boldsymbol{\alpha}}. \tag{22}$$

By suppressing the objects in upper-scripts for simplicity, with the notation $W_t^{\mathrm{u}} = W_t^{\mathrm{u},\boldsymbol{\mu},\boldsymbol{\alpha}}$ and $\mathbf{X}_{\mathrm{u}}(t) = \bar{\mathbf{X}}_{\mathrm{u}}(t)$, one can recover the original mean-field forward SDE defined in Eq (1). Note that this formulation reveals that the process $\bar{\mathbf{X}}_{\mathrm{u}}(t)$ is a weak solution under $\mathbb{P}^{\boldsymbol{\mu},\boldsymbol{\alpha}}$, and the cost functional can be posed as follows:

$$\mathcal{J}(\boldsymbol{\nu}^{\boldsymbol{\alpha}}, \boldsymbol{\alpha}) = \int_{\mathbb{T}} \mathbb{E}_{\boldsymbol{\alpha},\boldsymbol{\nu}} \left[ \|\mathbb{E}_{\mathrm{u}\sim p(\mathrm{u})} \mathbf{X}_{\mathrm{u}}^{\boldsymbol{\alpha}}(t) - y_t\|_E^2 + \mathbf{G}(\mathbf{X}_{\mathrm{u}}^{\boldsymbol{\alpha}}(T), \boldsymbol{\nu}^{\boldsymbol{\alpha}}) \right] dt, \tag{23}$$

where the expectation $\mathbb{E}_{\boldsymbol{\alpha},\boldsymbol{\nu}}$ is taken with respect to $\mathbb{P}^{\boldsymbol{\mu},\boldsymbol{\alpha}}$. Note that the cost functional in Eq. (3) is an alternative form of Eq. (23). Next, we reformulate the approximation of mean-field games with graphon in the probabilistic sense. Let $\boldsymbol{\alpha} = \alpha(t,\mathrm{x};\theta) := \hat{\alpha}(t,\mathrm{x},\bar{\mu},\bar{\mathfrak{e}}) := \hat{\boldsymbol{\alpha}}$ be an extended control with fixed arguments $\bar{\mu}, \bar{\mathfrak{e}}$. For the fixed $\boldsymbol{\nu}_u^{\boldsymbol{\alpha}^*}$ a.e., $\mathrm{u} \sim \mathrm{v}_{\mathrm{Unif}}$ associated with the optimal control $\hat{\boldsymbol{\alpha}}^*$, let us consider a Hamiltonian-Jacobi-Bellman equation (HJBE), having a classical value function $\mathcal{V}$:

$$\partial_t \mathcal{V}(t,\mathrm{x}) + \frac{1}{2}\mathbf{Tr}[\sigma_t^2 \partial_{\mathrm{xx}}^2 \mathcal{V}(t,\mathrm{x})] + H\left(t,\mathrm{x}, \boldsymbol{\nu}_u^{\boldsymbol{\alpha}^*}, \partial_{\mathrm{x}}\mathcal{V}(t,\mathrm{x}), \hat{\boldsymbol{\alpha}}^*(t,\mathrm{x}, \boldsymbol{\nu}_u^{\boldsymbol{\alpha}^*}, \partial_{\mathrm{x}}\mathcal{V}(t,\mathrm{x}))\right) = 0, \tag{24}$$

Then, forward-backward SDEs associated with the Hamiltonian system in (24) can be described in the Proposition A.1:

---

**Proposition A.1.** *(Weak Formulation: Forward-Backward SDEs I) (Carmona & Delarue, 2013) For the fixed flow of measures $\nu_u(\cdot) : \mathbb{T} \to \mathcal{P}_2$ and the fixed label $\mathrm{u}$, let $(\mathbf{X}_{\mathrm{u}}(t), \mathbf{Y}_{\mathrm{u}}(t), \mathbf{Z}_{\mathrm{u}}(t))$ be a family of processes that solves forward-backward stochastic differential equations with respect to the proposed graphon system in Eq (1) given as follows:*

$$d\mathbf{X}_{\mathrm{u}}(t) = \left(\boldsymbol{b}_W(\mathbf{X}_u(t), \nu_u, \hat{\boldsymbol{\alpha}}^*) + \boldsymbol{b}(t, \mathbf{X}_u(t), \hat{\boldsymbol{\alpha}}^*)\right) dt + \sigma_t dW_t^{\mathrm{u}}, \tag{25}$$

$$d\mathbf{Y}_{\mathrm{u}}(t) = -H(t, \mathbf{X}_{\mathrm{u}}(t), \mathbf{Y}_{\mathrm{u}}(t), \nu_u, \hat{\boldsymbol{\alpha}}^*)dt + \mathbf{Z}_{\mathrm{u}}(t) \cdot dW_t^{\mathrm{u}}. \tag{26}$$

*where $\boldsymbol{b}_W(\mathbf{x}, \nu, \alpha) := \langle \mathbb{W}_{\boldsymbol{\alpha}}[\nu](\mathrm{u}), \boldsymbol{\psi}\rangle(\mathrm{x}, \alpha)$ is the graphon interaction term, and terminal constraint is given as $\mathbf{Y}_{\mathrm{u}}(T) = \mathbf{G}(\mathbf{X}_T, \boldsymbol{\nu}_T)$. Then, under the mild assumption (e.g., smooth boundness of $\partial_x \mathcal{V}$ and $\partial_{xx}\mathcal{V}$), there exist solutions of stochastic optimal control of the following minimization problem:*

$$\inf_{\boldsymbol{\alpha}\in\mathbb{A}} \mathcal{J}(\boldsymbol{\nu}^{\boldsymbol{\alpha}}, \boldsymbol{\alpha}) = \mathbf{Y}_{\mathrm{u}}(0). \tag{27}$$

---

For the closed Markovian control such as neural control introduced in Section 2, the solution to adjoint process $\mathbf{Z}_{\mathrm{u}}(t)$ can be defined as stated in Definition A.2. By rewriting forward-backward SDEs in Eq (25) and Eq (26) for non-optimal neural controls $\boldsymbol{\alpha}$ (*i.e.*, neural networks) which are updated via gradient descent, we can recover the proposed gradeint system of FBSDEs in Definition 3.2.

---

**Definition A.2.** *(Gradient System of FBSDEs). For the fixed flow of measures $\nu_u(\cdot) : \mathbb{T} \to \mathcal{P}_2$ and the fixed label $\mathrm{u}$ at each stage $m$, we consider a family of processes $(\mathbf{X}_{\mathrm{u}}(t), \mathbf{Y}_{\mathrm{u}}(t), \mathbf{Z}_{\mathrm{u}}(t))$ that solves forward-backward stochastic differential equations with respect to the proposed graphon system in Eq (1) given as follows:*

$$d\mathbf{X}_{\mathrm{u}}^{m,\boldsymbol{\alpha}_m}(t) = \boldsymbol{b}_W(\mathbf{X}_u^{m,\boldsymbol{\alpha}_m}(t), \nu_{\mathrm{u}}, \boldsymbol{\alpha}_m)dt + \boldsymbol{b}(t, \mathbf{X}_u^{m,\boldsymbol{\alpha}_m}(t), \boldsymbol{\alpha}_m)dt + \sigma_t dW_t^{\mathrm{u}}, \tag{28}$$

$$d\mathbf{Y}_{\mathrm{u}}^{m,\boldsymbol{\alpha}_m}(t) = -H(t, \mathbf{X}_{\mathrm{u}}^{m,\boldsymbol{\alpha}_m}(t), \mathbf{Y}_{\mathrm{u}}^{m,\boldsymbol{\alpha}_m}(t), \nu_{\mathrm{u}}, \boldsymbol{\alpha}_m)dt - \mathbf{Z}_t^m \cdot dW_t^{\mathrm{u}}, \tag{29}$$

$$\boldsymbol{\alpha}_{m+1} := \alpha\left(t, \mathbf{X}_{\mathrm{u}}^{m,\boldsymbol{\alpha}_m}; \theta^m - \mathbb{E}_{\mathbf{Y},t\leq T}\left[\gamma^m \nabla_\theta \mathbf{Y}_{\mathrm{u}}^{m,\boldsymbol{\alpha}_m}(t)\right]\right) \in \mathbb{A}, \tag{30}$$

$$\nu_{\mathrm{u}} = \mathbf{Law}(\mathbf{X}_u^{m-1,\boldsymbol{\alpha}_{m-1}^*}), \tag{31}$$

*where $\gamma^m > 0$ is a learning rate of gradient descent, and $\mathbb{A}$ is a set of admissible neural agents. Then, we have $(\mathbf{Y}_{\mathrm{u}}(t), \mathbf{Y}_{\mathrm{u}}(T), \mathbf{Z}_{\mathrm{u}}(t)) = (\mathcal{J}, \mathbf{G}, (\partial_{\mathrm{x}}\mathcal{J})\sigma_t^{-1})$.*

---

A.4.2 ANALYSIS ON STOCHASTIC OPTIMALITY AND CONVERGENCE

**Stochastic Optimality.** In the following, we introduce the second type of forward-backward SDEs, which is based on the principles of stochastic maximum principle:

**Proposition A.3.** *(Stochastic Maximum Principle: Forward-Backward SDEs II) (Bensoussan et al., 2013) For the fixed flow of measures $\nu_u(\cdot) : \mathbb{T} \to \mathcal{P}_2$ and the fixed label* u*, let* $(\mathbf{X}_u(t), \mathbf{Y}_u^{MP}(t), \mathbf{Z}_u^{MP}(t))$ *be a family of processes that solves forward-backward stochastic differential equations with respect to the proposed graphon system in Eq* (1) *given as follows:*

$$d\mathbf{X}_u(t) = \left(\boldsymbol{b}_W(\mathbf{X}_u(t), \nu_u, \hat{\boldsymbol{\alpha}}^*) + \boldsymbol{b}(t, \mathbf{X}_u(t), \hat{\boldsymbol{\alpha}}^*)\right) dt + \sigma_t dW_t^u,$$

$$d\mathbf{Y}_u^{MP}(t) = -\partial_x H(t, \mathbf{X}_u(t), \mathbf{Y}_u^{MP}(t), \nu_u, \hat{\boldsymbol{\alpha}}^*)dt + \mathbf{Z}_t^{MP} \cdot dW_t^u.$$

*For the progressively measurable admissible Markovian neural control $\boldsymbol{\beta}$ under the mild assumption (e.g., smooth boundness of $\partial_x \mathcal{V}$ and $\partial_{xx} \mathcal{V}$), there exists a constant $\tau_{SMP} > 0$ such that the following inequality holds:*

$$\mathcal{J}(\boldsymbol{\nu}^{\hat{\boldsymbol{\alpha}}^*}, \hat{\boldsymbol{\alpha}}^*) + \tau_{SMP} \int_{\mathbb{T}} \|\hat{\boldsymbol{\alpha}}^* - \boldsymbol{\beta}\|_{\boldsymbol{\nu}} dt \leq \mathcal{J}(\boldsymbol{\nu}^{\boldsymbol{\beta}}, \boldsymbol{\beta}). \tag{32}$$

*Remark.* Note that the backward dynamics $\mathbf{Y}_u^{\mathbf{MP}}$ differs from the original backward dynamics $\mathbf{Y}_u$ in Definition (A.2) as the dynamics is designed to be associated with *Pontryagin stochastic maximum principle*. This principle plays a central role in the proof of Proposition A.4, demonstrating the stochastic optimality of neural agents in the following section.

In what it follows, we demonstrate that the stochastic optimality of the proposed gradient system can be guaranteed under the specific conditions required for constructing the control set in Prop A.4.

**Proposition A.4.** *(Maximum Principle of Graphon Mean-field System) Assume that there exists a constant $K_H$ such that $\|\partial_{\boldsymbol{\alpha}}\|\boldsymbol{H}\|_E\|_{\infty,\boldsymbol{\nu}} \leq K_H$. Then, there exists a convex set of admissible neural agents $\boldsymbol{\alpha}_m \in \mathbb{A}$ such that the following relation holds:*

$$D_{\boldsymbol{\alpha}} \mathcal{J}(\boldsymbol{\nu}^{\boldsymbol{\alpha}_m}, \boldsymbol{\alpha}_m) := \lim_{\varepsilon \to 0} \frac{d}{d\varepsilon} \mathcal{J}\left[\boldsymbol{\alpha}_m + \varepsilon(\boldsymbol{\alpha}_m - \boldsymbol{\alpha}_{m-1})\right] \xrightarrow{m \to \infty} 0. \tag{33}$$

*Furthermore, the sequence of control profile $\{\boldsymbol{\alpha}_m\}$ leads to the minimization of the stochastic Hamiltonian system in terms of **Pontryagin maximum principle**:*

$$\lim_{m \to \infty} H(t, \mathbf{X}_u^m(t), \mathbf{Y}_u^{m,MP}(t), \nu_u, \boldsymbol{\alpha}_m) = \inf_{\boldsymbol{\alpha} \in \mathbb{A}} H(t, \mathbf{X}_u(t), \mathbf{Y}_u^{MP}(t), \nu_u, \boldsymbol{\alpha}), \ dt \otimes d\mathbb{P} - a.e. \tag{34}$$

*where the population is set to $\nu_u = \Psi(\boldsymbol{\nu}^{\boldsymbol{\alpha}_{m-1}}) := \Psi(\nu_u^{\boldsymbol{\alpha}_{m-1}})$. In other words, the value function can be derived by the proposed gradient system of FBSDEs:*

$$\mathcal{V} := \inf_{\boldsymbol{\alpha} \in \mathbb{A}} \mathcal{J}(\boldsymbol{\nu}^{\boldsymbol{\alpha}}, \boldsymbol{\alpha}) = \lim_{m \to \infty} \mathcal{J}(\boldsymbol{\nu}^{\boldsymbol{\alpha}_m}, \boldsymbol{\alpha}_m). \tag{35}$$

*Proof.* We divide the proof into two separate steps.

**1. Computation of Gâteaux derivative $D_{\boldsymbol{\alpha}} \mathcal{J}$.** The aim of the first step is to provide an explicit computation of the Gâteaux derivative of cost functional (value function) with respect to the neural agent. To achieve this, we introduce the variation equation $i_u$ and its associated gradient system of SDEs with fixed $\boldsymbol{\beta}$:

$$d\mathbf{Y}_u^{m,\mathbf{MP}}(t) = -\partial_x H(t, \mathbf{X}_u^m(t), \mathbf{Y}_u^{m,\mathbf{MP}}(t), \nu_u, \hat{\boldsymbol{\alpha}}_m)dt + \mathbf{Z}_t^{m,\mathbf{MP}} \cdot dW_t^u, \tag{36}$$

$$di_u(t) = [(\partial_x \boldsymbol{b}_W + \partial_x \boldsymbol{b})i_u(t)]dt + [(\partial_{\boldsymbol{\alpha}} \boldsymbol{b}_W + \partial_{\boldsymbol{\alpha}} \boldsymbol{b})\boldsymbol{\beta}_m]dt, \tag{37}$$

$$dj_u(t) := d[i_u(t) \cdot \mathbf{Y}_u^{m,\mathbf{MP}}(t)]dt \in \mathbb{R}^d. \tag{38}$$

Let $\Upsilon_{\boldsymbol{\alpha}}(m, \epsilon) := \boldsymbol{\alpha}_m + \epsilon\boldsymbol{\beta}_m$ represent the infinitesimal changes of the admissible neural agent $\boldsymbol{\alpha}^m$ in the direction of $\boldsymbol{\beta}_m := \boldsymbol{\alpha}_{m-1} - \boldsymbol{\alpha}_m$. To feasibly select the convex combination $\Upsilon_{\boldsymbol{\alpha}}(m, \epsilon)$ for any

$m$ and $\epsilon \in [0, 1]$, both neural agents need to lie within some convex set $\mathbb{A}_m$. For now, we assume that there exists a convex set $\mathbb{A}_m$ that includes $\boldsymbol{\alpha}$ and $\boldsymbol{\beta}$. The explicit form of this set will be clarified in the subsequent step. Given the definition, we compute the derivative as follows:

$$
\begin{aligned}
D_{\boldsymbol{\alpha}} \mathcal{J}(\boldsymbol{\nu}^{\boldsymbol{\alpha}_m}, \boldsymbol{\alpha}_m) &= \frac{d}{d\epsilon} \mathcal{J}(\boldsymbol{\nu}^{\Upsilon_{\boldsymbol{\alpha}}(m,\epsilon)}, \Upsilon_{\boldsymbol{\alpha}}(m, \epsilon))|_{\epsilon=0} \\
&= \mathbb{E}\left[ \int_{\mathbb{T}} [\mathfrak{i}_{\mathrm{u}}(t) \partial_{\mathrm{x}} f + \boldsymbol{\beta}_m \partial_{\boldsymbol{\alpha}} f] dt + \mathfrak{i}_{\mathrm{u}}(T) \partial_x \boldsymbol{G} \right],
\end{aligned}
\tag{39}
$$

where we denote $f(t, \mathrm{x}, \alpha) = \|\mathbb{E}_{\mathrm{u}}[\mathrm{x}^{\alpha}(t)] - y_t\|^2$. While $\mathfrak{i}_{\mathrm{u}}(T) \partial_x \boldsymbol{G}$ can be identified with $\mathfrak{j}_{\mathrm{u}}(T)$, we apply the product rule to the third dynamics $d\mathfrak{j}_{\mathrm{u}}$ in Eq (38) to have variational form to induce $\mathfrak{j}_{\mathrm{u}}(T)$:

$$
\begin{aligned}
d\mathfrak{j}_{\mathrm{u}}(t) &= [\mathbf{Y}_{\mathrm{u}}(t) \cdot d\mathfrak{i}_{\mathrm{u}}(t)] dt + [\mathfrak{i}_{\mathrm{u}}(t) \cdot d\mathbf{Y}_{\mathrm{u}}^{\mathbf{MP}}(t)] dt + \mathbf{Tr}[d\mathbf{Y}_{\mathrm{u}}^{\mathbf{MP}}(t) \otimes d\mathfrak{i}_{\mathrm{u}}(t)] \\
&= \int_0^T \mathbf{Y}_{\mathrm{u}}^{\mathbf{MP}}(t) \cdot (\partial_{\mathrm{x}} \boldsymbol{b}_W + \partial_{\mathrm{x}} \boldsymbol{b}) \boldsymbol{\beta}_m + \mathbf{Y}_{\mathrm{u}}^{\mathbf{MP}}(t) \cdot (\partial_{\boldsymbol{\alpha}} \boldsymbol{b}_W + \partial_{\boldsymbol{\alpha}} \boldsymbol{b}) \mathfrak{i}_{\mathrm{u}}(t) dt \\
&= \int_0^T \partial_{\mathrm{x}} \boldsymbol{G} \cdot (\partial_{\mathrm{x}} \boldsymbol{b}_W + \partial_{\mathrm{x}} \boldsymbol{b}) \boldsymbol{\beta}_m + \partial_{\mathrm{x}} \boldsymbol{G} \cdot (\partial_{\boldsymbol{\alpha}} \boldsymbol{b}_W + \partial_{\boldsymbol{\alpha}} \boldsymbol{b}) \mathfrak{i}_{\mathrm{u}}(t) dt.
\end{aligned}
\tag{40}
$$

Combining Eq (39) with Eq (40), and Cauchy–Schwarz inequality gives explicit form for the Gâteaux derivative of objective functional.

$$
\begin{aligned}
D_{\boldsymbol{\alpha}} \mathcal{J}(\Upsilon_{\boldsymbol{\alpha}}(m, \epsilon)) &= \mathbb{E}\left[ \int_{\mathbb{T}} \partial_{\boldsymbol{\alpha}} H(t, \mathbf{X}_{\mathrm{u}}^m, \mathbf{Y}_{\mathrm{u}}^{m,\mathbf{MP}}(t), \Psi(\boldsymbol{\nu}^{\boldsymbol{\alpha}_{m-1}}), \boldsymbol{\alpha}_m) dt \cdot \boldsymbol{\beta}_m \right] \\
&\le \mathbb{E}\left[ \int_{\mathbb{T}} \|\partial_{\boldsymbol{\alpha}} H(t, \mathbf{X}_{\mathrm{u}}^m, \mathbf{Y}_{\mathrm{u}}^{m,\mathbf{MP}}(t), \Psi(\boldsymbol{\nu}^{\boldsymbol{\alpha}_{m-1}}), \boldsymbol{\alpha}_m)\|_E \cdot \|\boldsymbol{\beta}_m\|_E dt \right] \\
&\le \left| \left| \|\partial_{\boldsymbol{\alpha}} \mathbf{H}^m\|_E \right| \right|_{\infty} \cdot \left| \left| \|\boldsymbol{\beta}_m\|_E \right| \right|_1,
\end{aligned}
\tag{41}
$$

where $\|\cdot\|_p$ denotes $L_p$-norm, and the last inequality is obtained by applying Hölder's inequality with the conjugate pair $(p = \infty, q = 1)$. Then, we have

$$
\begin{aligned}
\left| \left| \|\boldsymbol{\beta}_m\|_E \right| \right|_1 &= \left| \left| \|\boldsymbol{\alpha}_m - \boldsymbol{\alpha}_{m-1}\|_E \right| \right|_1 := \left| \left| \|\alpha(t, \mathbf{X}_{\mathrm{u}}^m, \theta^m) - \alpha(t, \mathbf{X}_{\mathrm{u}}^m, \theta^{m-1})\|_E \right| \right|_1 \\
&\le \gamma^{m-1} \mathbf{Lip}_{\alpha} \mathbb{E} \delta_{\theta} \mathbf{Y}^{m-1}, \qquad \delta_{\theta} \mathbf{Y}^{m-1} := \|\nabla_{\theta} \mathbf{Y}_{\mathrm{u}}^{m-1, \boldsymbol{\alpha}_{m-1}}(t)\|_E.
\end{aligned}
\tag{42}
$$

**2. Construction of $\mathbb{A}$.** Next, we define the explicit form of the control set $\mathbb{A}_m$. The constructed control set must meet two conditions: (1) it must be convex, and (2) the right-hand side of the inequality in Eq (41) must converge. For properly dealing with the first condition, let us consider a metric ball $\mathbf{B}_m$ in $L_1$ space as follows:

$$
\mathbf{B}_m := B(\boldsymbol{\alpha}_{m-1}, \boldsymbol{r}_m) \in \mathbb{L}_1,
\tag{43}
$$

$$
\boldsymbol{r}_m := r_{\mathrm{u},t,m} = \varepsilon \gamma^{m-1} \mathbf{Lip}_{\alpha} \delta_{\theta} Y_{\mathrm{u}}^{m-1}(t), \quad \varepsilon \in [0, 1].
\tag{44}
$$

Since any arbitrary metric ball is convex and the calculated reverse direction of gradient guarantees local minimum at each stage, the setup of the proposed metric ball ensures the well-definedness of Gâteaux derivative in Eq (41) and local optimality at each stage $m$.

Let $\lambda_{\max}^m(\boldsymbol{\alpha})$ be an eigenvalue with respect to the principal direction of Hessian for cost functional, $i.e.$, $\mathbf{Hess}_{\theta} \mathcal{J}(\boldsymbol{\nu}^{\boldsymbol{\alpha}}, \boldsymbol{\alpha}(\cdot; \theta))$. Consider another control set $\mathbb{C}_m := \{\boldsymbol{\alpha}_{m-1}; \lambda_{\max}^{m-1}(\boldsymbol{\alpha}_{m-1}) \le (\gamma^{m-1})^{-1}\}$. The conventional analysis of gradient descent gives the following inequality on $\mathbb{C}_m$:

$$
\mathbb{E}\mathbf{Y}^{m,\boldsymbol{\alpha}_m} \le \mathbb{E}\mathbf{Y}^{m-1,\boldsymbol{\alpha}_{m-1}} - \frac{1}{2}\left(2\gamma^{m-1} - (\gamma^{m-1})^2 \lambda_{\max}^{m-1}(\boldsymbol{\alpha}_{m-1})\right)(\mathbb{E}\delta_{\theta}\mathbf{Y}^{m-1})^2.
\tag{45}
$$

While the second term in right-hand side of Eq (45) is non-negative, the sequence of expectations for the backward dynamics is non-increasing, demonstrating that $\lim_{m\to\infty} D_{\boldsymbol{\alpha}}\mathcal{J} \le \lim_{m\to\infty} \mathbb{E}\delta_{\theta}\mathbf{Y}^{m-1} = 0$ when the infinite sequence $\{\boldsymbol{\alpha}_m\}$ lies within $\lim_{m\to\infty} \mathbb{C}_m$. To inherit aforementioned properties lying in both control profiles for all $m$, we define $\mathbb{A}_m := \bigsqcup_{m \ge \mathfrak{m}} (\mathbb{B}_{\mathfrak{m}} \cap \mathbb{C}_{\mathfrak{m}})$, where $\mathbb{A} = \lim_{m\to\infty} \mathbb{A}_m$. The result directly follows from findings in the stochastic maximum

principle (SMP) (Carmona et al., 2018; Bensoussan et al., 2013), ensuring the equivalence of the following relation:

$$\mathbb{E}\partial_{\boldsymbol{\alpha}}\mathbf{H}(\cdot, \boldsymbol{\alpha}^*) \cdot \boldsymbol{\beta}_m = 0 \quad \longleftrightarrow \quad \boldsymbol{\alpha}^* = \arg\inf_{\boldsymbol{\alpha} \in \mathbb{A}} \mathbf{H}(\cdot, \boldsymbol{\alpha}). \tag{46}$$

Note that this equivalence relation is applicable only when $\mathbb{A}$ is constructed in the manner previously specified. $\qquad\square$

### A.4.3 CONVERGENCE OF GRADIENT SYSTEM OF FBSDES, MEAN-FIELD EQUILIBRIUM

As we have formally defined the stochastic optimal control problem and established the corresponding optimality conditions, this section delves into the detailed rationale of how the proposed gradient descent-based FBSDEs achieve the Nash equilibrium. We will prove Proposition 3.5 through the following steps:

1. For the arbitrary probability measures (*i.e.*, $\boldsymbol{\mu}^{\boldsymbol{\beta}}, \boldsymbol{\nu}^{\boldsymbol{\alpha}}$) associated with fixed Markovian controls $\boldsymbol{\alpha}$ and $\boldsymbol{\beta}$, we first establish that the upper bounds of the generalized Wasserstein distance remain stable when two measure-valued operators $\Phi$ and $\Psi$ are composed repeatedly:

$$\mathcal{W}_{t,\mathcal{M}}([\Phi \circ \Psi]^{\circ m}(\boldsymbol{\mu}^{\boldsymbol{\beta}}), [\Phi \circ \Psi]^{\circ m}(\boldsymbol{\nu}^{\boldsymbol{\alpha}})) \xrightarrow{m \geq M} 0. \tag{47}$$

2. Consequently, we reparameterize reference measures $(\boldsymbol{\mu}^{\boldsymbol{\beta}}, \boldsymbol{\nu}^{\boldsymbol{\alpha}})$ with the laws of inferred mean-field forward dynamics in Eq 1 at subsequent stages (*i.e.*, $\boldsymbol{\nu}^{\boldsymbol{\alpha}^m}, \boldsymbol{\nu}^{\boldsymbol{\alpha}^{m+1}}$), proving the convergence towards mean-field Nash equilibrium.

**Proposition 3.5.** *With the assumptions explored in the previous proof, for the fixed label* $\mathrm{u} \sim p(\mathrm{u})$, *the* $m$-*fold of composition* $\Phi \circ \Psi$ *induces convergent behavior of generalized Wasserstein distance:*

$$\mathcal{W}_2([\Phi \circ \Psi]^{\circ m}(\boldsymbol{\nu}^{\boldsymbol{\alpha}_1}), [\Phi \circ \Psi]^{\circ m}(\boldsymbol{\nu}^{\boldsymbol{\alpha}_0}))^2 \leq \sup_{t \in \mathbb{T}} \mathcal{W}_{t,\mathcal{M}}([\Phi \circ \Psi]^{\circ m}(\boldsymbol{\nu}^{\boldsymbol{\alpha}_1}), [\Phi \circ \Psi]^{\circ m}(\boldsymbol{\nu}^{\boldsymbol{\alpha}_0}))^2$$

$$\leq \lim_{M \to \infty} \frac{C(T)^M (\sup_t \sup_m \mathbf{r}_m)^M - 1}{C(T)(\sup_t \sup_m \mathbf{r}_m) - 1} + \frac{(C'T)^M}{M!} \sup_{t \in \mathbb{T}} \mathcal{W}_{t,\mathcal{M}}(\boldsymbol{\nu}^{\boldsymbol{\alpha}_1}, \boldsymbol{\nu}^{\boldsymbol{\alpha}_0})^2 \xrightarrow{M \to \infty} 0. \tag{48}$$

*where a numerical constant* $C$ *is dependent on* $b_0, C_1, H_{\psi}, \mathbf{Lip_b}, \mathfrak{m}_2, |\mathbb{O}|, e^{-|\mathbb{O}|}, \mathfrak{h}, \mathbf{Lip}_W$. *In other words,* $[\Phi \circ \Psi]^{\circ m}$ *is a Cauchy sequence on* $\mathcal{M}$, *and the proposed gradient system converges.*

*Proof.* Recall the definition of controlled graphon system that the particle dynamics at time $t$ with distinctive controls $\boldsymbol{\alpha}$ and $\boldsymbol{\beta}$ can be presented as follows:

$$\mathbf{X}_{\mathrm{u}}^{\boldsymbol{\nu}, \boldsymbol{\alpha}}(t) = \mathbf{X}_{\mathrm{u}}^{\boldsymbol{\nu}, \boldsymbol{\alpha}}(0) + \int_0^t \langle \mathbb{W}_{\boldsymbol{\alpha}}[\nu_{\mathrm{v},s}], \boldsymbol{\psi} \rangle (\mathbf{X}_{\mathrm{u}}^{\boldsymbol{\alpha}}(s)) ds + \int_0^t \boldsymbol{b}(s, \mathbf{X}_u^{\boldsymbol{\alpha}}(s), \boldsymbol{\alpha}) ds + \int_0^t \sigma_s dW_s^{\mathrm{u}},$$

$$\mathbf{X}_{\mathrm{u}}^{\boldsymbol{\mu}, \boldsymbol{\beta}}(t) = \mathbf{X}_{\mathrm{u}}^{\boldsymbol{\mu}, \boldsymbol{\beta}}(0) + \int_0^t \langle \mathbb{W}_{\boldsymbol{\beta}}[\mu_{\mathrm{v},s}], \boldsymbol{\psi} \rangle (\mathbf{X}_{\mathrm{u}}^{\boldsymbol{\beta}}(s)) ds + \int_0^t \boldsymbol{b}(s, \mathbf{X}_u^{\boldsymbol{\beta}}(s), \boldsymbol{\beta}) ds + \int_0^t \sigma_s dW_s^{\mathrm{u}}.$$

Given the dynamics above, the property of measure projection $\Psi$ induces the upper bound of generalized Wasserstein distance as follows:

$$\mathcal{W}_{t,\mathcal{M}}(\Phi(\boldsymbol{\mu}^{\boldsymbol{\beta}}), \Phi(\boldsymbol{\nu}^{\boldsymbol{\alpha}}))^2 \leq \mathbb{E}\left[\sup_{s \leq t} \|\mathbf{X}_u^{\boldsymbol{\mu}, \boldsymbol{\beta}}(s) - \mathbf{X}_u^{\boldsymbol{\nu}, \boldsymbol{\alpha}}(s)\|^2\right]$$

$$\leq b_0 \mathbb{E}\left[\int_0^t \int_{\mathbb{O}} \|\int_{\mathbb{R}^d} \boldsymbol{\psi}(\mathbf{X}_{\mathrm{u}}^{\boldsymbol{\mu}, \boldsymbol{\beta}}(s), \mathbf{Y}) W_{\boldsymbol{\beta}}(\mathrm{u}, \mathrm{v}) d\mu_{\mathrm{v},s}(\mathbf{Y}) \right.$$

$$\left. - \int_{\mathbb{R}^d} \boldsymbol{\psi}(\mathbf{X}_{\mathrm{u}}^{\boldsymbol{\nu}, \boldsymbol{\alpha}}(s), \hat{\mathbf{Y}}) W_{\boldsymbol{\alpha}}(\mathrm{u}, \mathrm{v}) d\nu_{\mathrm{v},s}(\hat{\mathbf{Y}})\|^2 d\mathrm{v}_{\mathrm{Unif}} ds\right]$$

$$+ b_0 \mathbb{E}\left[\int_0^t \|\boldsymbol{b}(s, \mathbf{X}_u^{\boldsymbol{\mu}, \boldsymbol{\alpha}}(s), \boldsymbol{\alpha}) - \boldsymbol{b}(s, \mathbf{X}_u^{\boldsymbol{\nu}, \boldsymbol{\beta}}(t), \boldsymbol{\beta})\|^2 ds\right]$$

$$\leq 3b_0 (\mathrm{I} + \mathrm{II} + \mathrm{III}) + b_0 \mathrm{IV},$$

$$\tag{49}$$

where the first and second inequalities are induced from Holder's inequality and the Burkholder-Davis-Gundy (Chaintron & Diez, 2022) with some constant $b_0 > 0$. Following the assumptions in Section A.3 and the modeling of graphons in Section 2, the first term (*i.e.*, I) can be upper-bounded in the following estimation.

$$
\begin{aligned}
\mathrm{I} &:= \mathbb{E}\left[\int_0^t \int_\mathbb{O} \left\|\int_{\mathbb{R}^d} \left[\boldsymbol{\psi}(\mathbf{X}_\mathrm{u}^{\boldsymbol{\nu},\boldsymbol{\alpha}}(s),\hat{\mathbf{Y}}) - \boldsymbol{\psi}(\mathbf{X}_\mathrm{u}^{\boldsymbol{\mu},\boldsymbol{\beta}}(s),\hat{\mathbf{Y}})\right] W_{\boldsymbol{\alpha}}(\mathrm{u},\mathrm{v}) d\nu_{\mathrm{v},s}(\hat{\mathbf{Y}})\right\|^2 d\mathrm{v}_{\mathrm{Unif}} ds\right] \\
&\le \mathbf{Lip}(\boldsymbol{\psi})^2 \mathbb{E}\left[\int_0^t \int_\mathbb{O} W_{\boldsymbol{\alpha}}^2(\mathrm{u},\mathrm{v}) \int_{\mathbb{R}^d} \left\|\mathbf{X}_\mathrm{u}^{\boldsymbol{\nu},\boldsymbol{\alpha}}(s) - \mathbf{X}_\mathrm{u}^{\boldsymbol{\nu},\boldsymbol{\beta}}(s)\right\|^2 d\nu_{\mathrm{v},s}(\hat{\mathbf{Y}}) d\mathrm{v}_{\mathrm{Unif}} ds\right].
\end{aligned}
\tag{50}
$$

Given the fixed control $\boldsymbol{\alpha} = \bar{\boldsymbol{\alpha}}$, optimizing the last inequality requires estimating the (local) Lipschitz continuity of positional encoding $\boldsymbol{\psi}$:

$$
\begin{aligned}
\mathbf{Lip}(\boldsymbol{\psi}(\cdot,\hat{\mathbf{Y}})) &\le \sup_{\mathrm{x}\in\mathbb{R}^d\setminus\{\hat{\mathbf{Y}}\}} \|\nabla\boldsymbol{\psi}(\mathrm{x},\hat{\mathbf{Y}})\| \\
&\le H_{\boldsymbol{\psi}}(\bar{\boldsymbol{\alpha}}) \sup_{\mathrm{x}\in\mathbb{R}^d\setminus\{\hat{\mathbf{Y}}\}} \mathfrak{a}^{-2}\left\|\left(\mathbf{I}_d - \frac{2(\mathrm{x}-\hat{\mathbf{Y}})\otimes_E(\mathrm{x}-\hat{\mathbf{Y}})}{\mathfrak{a}^2}\right)\right\|.
\end{aligned}
\tag{51}
$$

where $\mathfrak{a} = \|\mathrm{x} - \hat{\mathbf{Y}}\|$ and $\otimes_E$ denotes the Euclidean outer product. Following by the assumption (**H2**), Grönwall's inequality with the fact that $\mathbf{spec}(\nabla\boldsymbol{\psi}) := \lambda_1 \le \max(1,-1)\mathfrak{a}^{-2}$, we have

$$
\mathrm{I} \le C_1^2 H_{\boldsymbol{\psi}}^2(\bar{\boldsymbol{\alpha}})\mathfrak{h}(\boldsymbol{\beta})\mathbb{E}\left[\int_0^t \sup_{r\le s}\left\|\mathbf{X}_\mathrm{u}^{\boldsymbol{\nu},\boldsymbol{\alpha}}(r) - \mathbf{X}_\mathrm{u}^{\boldsymbol{\nu},\boldsymbol{\beta}}(r)\right\|^2 ds\right].
\tag{52}
$$

Since each component $\boldsymbol{\psi}_i$ possesses the same spectral norm as $\boldsymbol{\psi}$, the second term can be upper-bounded with the improved definition of generalized Wasserstein distance in Eq (9):

$$
\begin{aligned}
\mathrm{II} &:= \mathbb{E}\left[\int_0^t \int_\mathbb{O} \left\|\int_{\mathbb{R}^d} \boldsymbol{\psi}(\mathbf{X}_\mathrm{u}^{\boldsymbol{\mu},\boldsymbol{\beta}}(s),\hat{\mathbf{Y}})W_{\boldsymbol{\beta}}(\mathrm{u},\mathrm{v})d[\nu_{\mathrm{v},s} - \mu_{\mathrm{v},s}](\hat{\mathbf{Y}})\right\|^2 d\mathrm{v}_{\mathrm{Unif}} ds\right] \\
&\le d|\mathbb{O}|C_1^2\mathbb{E}\left[\sup_{\mathrm{u}\in\mathbb{O}} \max_{i\in\{1,\cdots,d\}} \int_0^t \left|\int_{\mathbb{R}^d} \frac{\boldsymbol{\psi}_i}{C_1}(\mathbf{X}_\mathrm{u}^{\boldsymbol{\mu},\boldsymbol{\beta}}(s),\cdot)W_{\boldsymbol{\beta}}(\mathrm{u},\mathrm{v})d[\nu_{\mathrm{v},s} - \mu_{\mathrm{v},s}]\right|^2 ds\right] \\
&\le d|\mathbb{O}|C_1^2\mathfrak{h}(\boldsymbol{\beta})\int_0^t \mathcal{W}_{s,\mathcal{M}}(\boldsymbol{\mu}^{\boldsymbol{\beta}},\boldsymbol{\nu}^{\boldsymbol{\alpha}})^2 ds.
\end{aligned}
\tag{53}
$$

Regarding the third term (*i.e.*, III), we have

$$
\begin{aligned}
\mathrm{III} &:= \mathbb{E}\left[\int_0^t \int_\mathbb{O} \left|\left|\int_{\mathbb{R}^d} \boldsymbol{\psi}(\mathbf{X}_\mathrm{u}^{\boldsymbol{\mu},\boldsymbol{\beta}}(s),\hat{\mathbf{Y}})|W_{\boldsymbol{\beta}} - W_{\boldsymbol{\alpha}}|d\nu_{\boldsymbol{v},s}(\hat{\mathbf{Y}})\right|\right|^2 d\mathrm{v}_{\mathrm{Unif}} ds\right] \\
&\le (2C_1^2\mathfrak{m}_2 H_{\boldsymbol{\psi}} + 1)\int_0^t \int_{\mathbb{O}^2} |W_{\boldsymbol{\beta}} - W_{\boldsymbol{\alpha}}|^2 d\mathrm{v}_{\mathrm{Unif}}^{\otimes 2}(\mathrm{u},\mathrm{v})ds \\
&\le (2C_1^2\mathfrak{m}_2 H_{\boldsymbol{\psi}} + 1)|\mathbb{T}|d_{\mathfrak{g}}^2(W_{\boldsymbol{\beta}},W_{\boldsymbol{\alpha}}).
\end{aligned}
\tag{54}
$$

The upper-bound of last term can be directly obtained by the Lipschitz condition.

$$
\begin{aligned}
\mathrm{IV} &:= \mathbb{E}\left[\int_0^t \|\boldsymbol{b}(s,\mathbf{X}_u^{\boldsymbol{\mu},\boldsymbol{\alpha}}(s),\boldsymbol{\alpha}) - \boldsymbol{b}(s,\mathbf{X}_u^{\boldsymbol{\nu},\boldsymbol{\beta}}(t),\boldsymbol{\beta})\|^2 ds\right] \\
&\le \mathbf{Lip}_{\boldsymbol{b}}\mathbb{E}\left[\int_0^t \sup_{r\le s}\left\|\mathbf{X}_u^{\boldsymbol{\mu},\boldsymbol{\alpha}}(r) - \mathbf{X}_u^{\boldsymbol{\nu},\boldsymbol{\beta}}(r)\right\|^2 ds\right] + \mathbf{Lip}_{\boldsymbol{b}}\int_0^t \sup_{r\le s}\|\boldsymbol{\alpha} - \boldsymbol{\beta}\|_{\nu_u(s)}^2 ds.
\end{aligned}
\tag{55}
$$

By replacing each term with numerical constants $C_3, C_4, C_5$ in the aggregation of all four terms, we finally have the following upper-bounds related to $d_{\mathfrak{g}}$, $\mathcal{W}_{\mathcal{M}}$ and $L_2$-norm:

$$
\mathbb{E}\left[\sup_{s\leq t}\|\mathbf{X}_u^{\boldsymbol{\mu},\boldsymbol{\beta}}(s) - \mathbf{X}_u^{\boldsymbol{\nu},\boldsymbol{\alpha}}(s)\|^2\right] \leq 3b_0\,(\mathrm{I+II+III}) + b_0(\mathrm{IV})
$$

$$
\leq \underbrace{b_0(3C_1 H_{\psi}(\bar{\boldsymbol{\alpha}})\mathfrak{h}(\boldsymbol{\beta}) + \mathbf{Lip}_{\boldsymbol{b}})}_{:=\log(C_3/t)}\mathbb{E}\left[\int_0^s \sup_{r\leq s}\left|\left|\mathbf{X}_{\mathrm{u}}^{\boldsymbol{\nu},\boldsymbol{\alpha}}(r) - \mathbf{X}_{\mathrm{u}}^{\boldsymbol{\nu},\boldsymbol{\beta}}(r)\right|\right|^2 dr\right]
$$

$$
+ \underbrace{(6b_0 C_1^2\mathfrak{m}_2 H_{\psi} + 3b_0)|\mathbb{T}|}_{:=C_4}\,d_{\mathfrak{g}}^2(W_{\boldsymbol{\beta}}, W_{\boldsymbol{\alpha}})
$$

$$
+ \underbrace{\max(3b_0 d|\mathbb{O}|C_1\mathfrak{h}(\boldsymbol{\beta}), \mathbf{Lip}_{\boldsymbol{b}})}_{:=C_5}\left(\int_0^t \sup_{r\leq s}\|\boldsymbol{\alpha}-\boldsymbol{\beta}\|_{\nu_{\mathrm{u}}(r)}^2 + \mathcal{W}_{s,\mathcal{M}}(\boldsymbol{\mu}^{\boldsymbol{\beta}},\boldsymbol{\nu}^{\boldsymbol{\alpha}})^2 ds\right). \quad (56)
$$

Applying Grönwall's inequality to the above result in Eq (56) and the first inequality in Eq (49) shows that there exists a constant $C' = 3\max(C_3, C_4, C_5)$ such that

$$
\mathcal{W}_{t,\mathcal{M}}(\Phi(\boldsymbol{\mu}^{\boldsymbol{\beta}}), \Phi(\boldsymbol{\nu}^{\boldsymbol{\alpha}}))^2 \leq C'\left(d_{\mathfrak{g}}^2(W_{\boldsymbol{\beta}}, W_{\boldsymbol{\alpha}}) + \int_0^t \sup_{r\leq s}\|\boldsymbol{\alpha}-\boldsymbol{\beta}\|_{\nu_{\mathrm{u}}(r)}^2 + \mathcal{W}_{s,\mathcal{M}}(\boldsymbol{\mu}^{\boldsymbol{\beta}},\boldsymbol{\nu}^{\boldsymbol{\alpha}})^2 ds\right).
$$
$$(57)$$

Next, the aim is to show the upper-bound of $d_{\mathfrak{g}}^2$, $\|\boldsymbol{\alpha}-\boldsymbol{\beta}\|_{\boldsymbol{\nu}}^2$ and $\mathcal{W}_{\cdot,\mathcal{M}}$. To proceed, let us first examine the upper bounds of the cut norms for both exponential and cosinusoidal graphons as follows:

$$
\sup_{A,B}\left|\int_{A\times B} W_{\boldsymbol{\alpha}}(\mathrm{u},\mathrm{v})dudv\right|^2 \leq \mathfrak{h}(\boldsymbol{\alpha}) = \begin{cases} W_0^2 + 2W_0(W_{1,l}+W_{2,l}) + (2/L)(\sum_l W_{1,l} + W_{2,l})^2 \\ (T/2)W_1^2\left(e^{-2T^{-1}|\mathbb{O}|} - 1\right). \end{cases}
$$
$$(58)$$

Modifying the upper-bound in Eq (58) by replacing $W_{\boldsymbol{\alpha}}$ with $\delta W := W_{\boldsymbol{\alpha}} - W_{\boldsymbol{\beta}}$, one can derive the following

$$
d_{\mathfrak{g}}^2(W_{\boldsymbol{\beta}}, W_{\boldsymbol{\alpha}}) = \|\delta W\|_{\mathfrak{g}}^2 \leq \max\left(11\mathbf{Lip}_W, (T/2)(e^{-2T^{-1}|\mathbb{O}|}-1)\right)\|\boldsymbol{\alpha}-\boldsymbol{\beta}\|_{\boldsymbol{\nu}}^2. \quad (59)
$$

At each stage $\{m\}_{1\leq m\leq M}$ with the given sequence of probability measures $\{\boldsymbol{\nu}^{\boldsymbol{\alpha}_m}\}_{1\leq m\leq M}$, we substitute $\Phi(\boldsymbol{\mu}^{\boldsymbol{\beta}})$ and $\Phi(\boldsymbol{\nu}^{\boldsymbol{\alpha}})$ in Eq (57) with $\Phi\circ\Psi(\boldsymbol{\nu}^{\boldsymbol{\alpha}_{m+1}})$ and $\Phi\circ\Psi(\boldsymbol{\nu}^{\boldsymbol{\alpha}_m})$, respectively. Then, one can derive the following relation:

$$
\mathcal{W}_{t,\mathcal{M}}(\Phi\circ\Psi(\boldsymbol{\nu}^{\boldsymbol{\alpha}_m}), \Phi\circ\Psi(\boldsymbol{\nu}^{\boldsymbol{\alpha}_{m-1}}))^2 = \mathcal{W}_{t,\mathcal{M}}(\mathbf{Law}(\mathbf{X}^{\boldsymbol{\nu}^*_{\boldsymbol{\alpha}_{m+1}},\boldsymbol{\alpha}^*_{m+1}}), \mathbf{Law}(\mathbf{X}^{\boldsymbol{\nu}^*_{\boldsymbol{\alpha}_m},\boldsymbol{\alpha}^*_m}))^2
$$

$$
\leq C'\left(d_{\mathfrak{g}}^2(W_{\boldsymbol{\alpha}^*_{m+1}}, W_{\boldsymbol{\alpha}^*_m}) + \int_0^t \sup_{r\leq s}\|\boldsymbol{\alpha}^*_{m+1}-\boldsymbol{\alpha}^*_m\|_{\nu_{\mathrm{u}}(r)} + \mathcal{W}_{s,\mathcal{M}}(\boldsymbol{\nu}^{\boldsymbol{\alpha}^*_{m+1}},\boldsymbol{\nu}^{\boldsymbol{\alpha}^*_m})^2 ds\right).
$$

$$
\leq C'\left(\max\left(t + 11\mathbf{Lip}_W, t + (T/2)(e^{-2T^{-1}|\mathbb{O}|}-1)\right)\right)\sup_t\|\alpha(t,\cdot,\theta^{m+1}) - \alpha(t,\cdot,\theta^m)\|_{\nu_{\mathrm{u}}(t)}^2
$$

$$
+ C'\int_0^t \mathcal{W}_{s,\mathcal{M}}(\boldsymbol{\nu}^{\boldsymbol{\alpha}^*_{m+1}},\boldsymbol{\nu}^{\boldsymbol{\alpha}^*_m})^2 ds
$$

$$
\leq \underbrace{C'\left(\max\left(t + 11\mathbf{Lip}_W, t + (T/2)(e^{-2T^{-1}|\mathbb{O}|}-1)\right)\right)}_{:=C(t)\,\leq\,C(T)}\left(\sup_t \boldsymbol{r}_m\right)
$$

$$
+ C'\int_0^t \mathcal{W}_{s,\mathcal{M}}(\boldsymbol{\nu}^{\boldsymbol{\alpha}^*_{m+1}},\boldsymbol{\nu}^{\boldsymbol{\alpha}^*_m})^2 ds,
$$

where the radius of metric ball (*i.e.*, $\mathbf{r}_m := r_{\mathrm{u},t,m}$) was defined in the proof of Proposition A.4. In the first equality, the controls $\boldsymbol{\alpha}$ are replaced with their optimal profiles $\boldsymbol{\alpha}^*$ following the definition of the operator $\Psi$ in. To set up the subsequent stage, we substitute a pair of controls $(\boldsymbol{\alpha}^*_{m+1}, \boldsymbol{\alpha}^*_m)$ with $(\boldsymbol{\alpha}_{m+1}, \boldsymbol{\alpha}_m)$ again. Next, we show the stability of the result obtained above for $M$-th stage by

observing the upper bound of $M$-fold of the operator composition.

$$\mathcal{W}_{t,\mathcal{M}}([\Phi \circ \Psi]^{\circ M}(\boldsymbol{\nu}^{\boldsymbol{\alpha}_1}), [\Phi \circ \Psi]^{\circ M}(\boldsymbol{\nu}^{\boldsymbol{\alpha}_0}))^2$$

$$\leq C(t) \sup_t \mathbf{r}_m + C' \int_0^t \mathcal{W}_{s_0,\mathcal{M}}([\Phi \circ \Psi]^{\circ M-1}(\boldsymbol{\nu}^{\boldsymbol{\alpha}_1}), [\Phi \circ \Psi]^{\circ M-1}(\boldsymbol{\nu}^{\boldsymbol{\alpha}_0}))^2 ds^0\Bigr)$$

$$\vdots \tag{60}$$

$$\leq \sum_{m=1}^M (C(t) \sup_t \mathbf{r}_m)^m + (C')^M \int_0^{s_0} \cdots \int_0^{s_M} \mathcal{W}_{s_m,\mathcal{M}}(\boldsymbol{\nu}^{\boldsymbol{\alpha}_{m+1}}, \boldsymbol{\nu}^{\boldsymbol{\alpha}_m})^2 d[\Pi^M](s^0, \cdots s^M).$$

where $d\Pi^m := ds^0 \otimes \cdots \otimes ds^m$ denotes $m$-product of Lebesgue measures $\{ds^m\}_{1 \leq m \leq M}$. Finally, we deduce that the supremum of the left-hand side can be controlled by

$$\lim_{M \to \infty} \sup_{t \in \mathbb{T}} \mathcal{W}_{t,\mathcal{M}}([\Phi \circ \Psi]^{\circ m}(\boldsymbol{\nu}^{\boldsymbol{\alpha}_1}), [\Phi \circ \Psi]^{\circ m}(\boldsymbol{\nu}^{\boldsymbol{\alpha}_0}))^2 \leq$$

$$+ \frac{C(T)^M (\sup_t \sup_m \mathbf{r}_m)^M - 1}{C(T)(\sup_t \sup_m \mathbf{r}_m) - 1} + \frac{(C'T)^M}{M!} \sup_{t \in \mathbb{T}} \mathcal{W}_{t,\mathcal{M}}(\boldsymbol{\nu}^{\boldsymbol{\alpha}_1}, \boldsymbol{\nu}^{\boldsymbol{\alpha}_0})^2 \longrightarrow 0. \tag{61}$$

where the learning rate $\gamma^m$ is chosen such that $\sup_t \sup_m \mathbf{r}_m \leq 1$ remains sufficiently small, and the last term in the inequality can be derived by modifying the following

$$\sup_{t \in \mathbb{T}} \mathcal{W}_{t,\mathcal{M}}(\Phi^{\circ m}(\nu^{\boldsymbol{\alpha}_1}), \Phi^{\circ m}(\nu^{\boldsymbol{\alpha}_0}))^2 \leq (C')^M \int_0^T \frac{(T-s)}{(m-1)!} \mathcal{W}_{s,\mathcal{M}}(\nu^{\boldsymbol{\alpha}_1}, \nu^{\boldsymbol{\alpha}_0})^2 ds. \tag{62}$$

The inequality in Eq (61) demonstrates that the sequence of operator compositions $\{[\Phi \circ \Psi]^{\circ m}\}_{m \leq M} : \mathcal{M} \to \mathcal{M}$ forms a Cauchy sequence, confirming the convergence of the proposed gradient system in the distributional sense. □

### A.4.4 Sampling Errors of Mean-field Predictors

Though not presented in the manuscript, the following result implies key theoretical conclusions: It demonstrates that the estimation errors for the neural agent, introduced by the sampled mean-field predictors (empirical measure) at the $m$-th gradient descent step, are kept within acceptable margins.

**Proposition A.5.** *(Worst-case Estimation Error of Neural Agents) Let $\mathbb{Q}_n := \mathbb{Q}_n(\mathrm{u}, t) = (1/n) \sum_i \delta_{\mathbf{X}_{\mathrm{u}_i}^{\boldsymbol{\alpha}}(t)}$ and $\mathbb{Q} := \nu_{\mathrm{u}}(t)$ be empirical laws of mean-field predictors and their corresponding mean-field limit. Then, the worst-case estimation error can be upper bounded with probability at least $1 - \delta$:*

$$\sup_{\boldsymbol{\alpha}_m \in \mathbb{A}} \left\| \int \boldsymbol{\alpha}^m d(\mathbb{Q}_n - \mathbb{Q}) \right\|_E^2 \leq \sqrt{\frac{32T^3(1+\mathfrak{m}_2)^2}{n} \ln\left(\frac{1}{\delta}\right)}$$

$$+ 4 \left( \sqrt{\frac{32}{n}} 2^{(3d-2)/2} \left( \varepsilon \gamma^{m-1} \mathbf{Lip}_\alpha \|\nabla_\theta \mathbf{Y}_{\mathrm{u}}^{m-1,\boldsymbol{\alpha}_{m-1}}(t)\|_E \right)^{d/2} \frac{d+2}{4(d-2)} \right)^{(d/2+2)^{-1}}. \tag{63}$$

**Remark.** While the admissible control set $\mathbb{A}$ guarantees the diminishing behavior of $\|\nabla_\theta \mathbf{Y}_{\mathrm{u}}^{m-1,\boldsymbol{\alpha}_{m-1}}(t)\|_E$, the second term in Eq (63) approaches zero as $m$ becomes large, even when $n$ is small.

*Proof.* The proof follows the standard convergence analysis of empirical processes. Let us fix the temporal variable $t$ and the labels of mean-field predictors $\mathrm{u}$. Then, one can show that the supremum of Euclidean norm can be decomposed as follows:

$$\sum_j^d \sup_{\pi_j \circ \boldsymbol{\alpha} \in \mathbb{A}_m^j} |\mathbb{E}_{\mathbb{Q}_n} \pi_j \circ \boldsymbol{\alpha}_m - \mathbb{E}_{\mathbb{Q}} \pi_j \circ \boldsymbol{\alpha}_m| \leq \sum_j^d \sup_{g \in \mathbb{A}_m^j} |\mathbb{E}_{\mathbb{Q}_n} g - \mathbb{E}_{\mathbb{Q}} g| := \boldsymbol{\Gamma}_{\mathbb{A}_m^j}(\mathbb{Q}_{\mathfrak{N}}, \mathbb{Q}), \tag{64}$$

where $\mathbf{\Gamma}_{\mathbb{A}_m^j}$ denotes the *integral probability metric* (Müller, 1997) with respect to the set $\mathbb{A}_m^j$ which consists of $j$-th component of neural agents at $m$-th stage. Note that the the supremum in the second term is taken for all function $g$ lying in the set of parameterized function, *i.e.*, neural agent. Let us define $\mathfrak{p}, \mathfrak{q} : (\mathbb{R}^d)^n \to \mathbb{R}$ such that $(\mathbf{X}_{\mathrm{u}_1}^{\boldsymbol{\alpha}}(t), \cdots, \mathbf{X}_{\mathrm{u}_n}^{\boldsymbol{\alpha}}(t)) \overset{\mathfrak{p}}{\mapsto} \sup_g |(1/n) \sum_i g(\mathbf{X}_{\mathrm{u}_i}^{\boldsymbol{\alpha}}(t)) - \mathbb{E}_{\mathbb{Q}} g|$, and $(\mathbf{X}_{\mathrm{u}_1}^{\boldsymbol{\alpha}}(t), \cdots, \mathbf{X}_{\mathrm{u}_n}^{\boldsymbol{\alpha}}(t)) \overset{\mathfrak{q}}{\mapsto} \mathbb{E}_{\boldsymbol{\sigma}} \sup_g |(1/n) \sum_i \sigma_i g(\mathbf{X}_{\mathrm{u}_i}^{\boldsymbol{\alpha}}(t))|$ where $\{\sigma_i\}_{i \leq n}$ is a set of i.i.d Rademacher random variables. Then both $\mathfrak{p}$ and $\mathfrak{q}$ satisfies the following inequality:

$$
\begin{aligned}
\sup_t \max_{i \in \{1, \cdots n\}} \Big| (\mathfrak{p}, \mathfrak{q})(\mathbf{X}_{\mathrm{u}_1}^{\boldsymbol{\alpha}}(t), \cdots, \mathbf{X}_{\mathrm{u}_{i-1}}^{\boldsymbol{\alpha}}(t), \mathrm{x}', \mathbf{X}_{\mathrm{u}_{i+1}}^{\boldsymbol{\alpha}}(t), \cdots \mathbf{X}_{\mathrm{u}_n}^{\boldsymbol{\alpha}}(t)) \\
- (\mathfrak{p}, \mathfrak{q})(\mathbf{X}_{\mathrm{u}_1}^{\boldsymbol{\alpha}}(t), \cdots, \mathbf{X}_{\mathrm{u}_n}^{\boldsymbol{\alpha}}(t)) \Big| \leq \frac{4T \sup_{\mathrm{x},t} \boldsymbol{\alpha}(t, \mathrm{x}; \theta)}{n}. \quad (65)
\end{aligned}
$$

Following by the McDiarmid's inequality with respect to $\mathfrak{p}$, we have two concentration inequalities:

$$
\exp\left( \frac{-n\varepsilon^2}{8T^2 \sup_{\mathrm{x},t} \boldsymbol{\alpha}(t, \mathrm{x}; \theta)^2} \right) \geq \quad \begin{cases} \mathbb{P}(\mathfrak{p} - \mathbb{E}\mathfrak{p} \geq \varepsilon) \\ \mathbb{P}(\mathfrak{q} - \mathbb{E}\mathfrak{q} \geq \varepsilon). \end{cases} \quad (66)
$$

By applying the symmetrization inequality (Wellner et al., 2013), we have the following inequality with probability at least $1 - \delta$

$$
\begin{aligned}
\mathbf{\Gamma}_{\mathbb{A}_m^j}(\mathbb{Q}_{\mathfrak{N}}, \mathbb{Q}) &\leq \mathbb{E}\mathbf{\Gamma}_{\mathbb{A}_m^j}(\mathbb{Q}_{\mathfrak{N}}, \mathbb{Q}) + \sqrt{\frac{8T^2 \sup_{\mathrm{x},t} \boldsymbol{\alpha}(t, \mathrm{x}; \theta)^2}{n} \ln\left(\frac{1}{\delta}\right)} \\
&\leq 2\tilde{\mathbb{E}}\mathbb{E}_{\boldsymbol{\sigma}} \left[ \sup_{g \in \mathbb{A}_m^j} \left| \frac{1}{n} \sum_i^n \sigma_i g(\mathbf{X}_{\mathrm{u}_i}^{\boldsymbol{\alpha}}(t)) \right| + \sqrt{\frac{8T^2 \sup_{\mathrm{x},t} \boldsymbol{\alpha}(t, \mathrm{x}; \theta)^2}{n} \ln\left(\frac{1}{\delta}\right)} \right] \\
&\leq 2\underbrace{\mathbb{E}_{\boldsymbol{\sigma}} \left[ \sup_{g \in \mathbb{A}_m^j} \left| \frac{1}{n} \sum_i^n \sigma_i g(\mathbf{X}_{\mathrm{u}_i}^{\boldsymbol{\alpha}}(t)) \right| \right]}_{\mathcal{R}_m(\mathbb{A}_m^j, \{\mathbf{X}_{\mathrm{u}_n}^{\boldsymbol{\alpha}}(t)\})} + \sqrt{\frac{32T^3(1 + \mathfrak{m}_2)^2}{n} \ln\left(\frac{1}{\delta}\right)}
\end{aligned} \quad (67)
$$

where the outer expectation is taken with respect to the randomness of mean-field predictors in the second line, and we apply McDiarmid's inequality in Eq (66) again to derive the last line. With the covering number of the Hilbert space for the $\mathbb{L}_2$-norm, we get

$$
\begin{aligned}
\mathcal{R}_m(\mathbb{A}_m^j, \{\mathbf{X}_{\mathrm{u}_n}^{\boldsymbol{\alpha}}(t)\}) &\leq \mathbb{E}_{\boldsymbol{\sigma}} \left[ \sup_{g \in \mathbb{A}_m^j} \left| \frac{1}{n} \sum_i^n \sigma_i g(\mathbf{X}_{\mathrm{u}_i}^{\boldsymbol{\alpha}}(t)) \right| \right] \\
&\leq \inf_{\epsilon > 0} \left\{ 2\epsilon + \sqrt{\frac{32}{n}} \int_{\epsilon/4}^{\infty} \sqrt{\mathcal{H}(\tau, \mathbb{A}_m^j, \mathbb{L}_2(\mathbb{Q}_n))} \right\} \\
&\leq \inf_{\epsilon > 0} \left\{ 2\epsilon + \sqrt{\frac{32}{n}} \int_{\epsilon/4}^{\infty} \left( \frac{2\mathbf{r}_m}{\tau} \right)^{d/2} d\tau \right\} \\
&\leq \inf_{\epsilon > 0} \left\{ 2\epsilon + \sqrt{\frac{32}{n}} (2\mathbf{r}_m)^{d/2} (\epsilon/4)^{-d/2+1} (d/2 - 1)^{-1} \right\} \\
&\leq \inf_{\epsilon > 0} \left\{ 2\epsilon + \sqrt{\frac{32}{n}} 2^{(3d-2)/2} \mathbf{r}_m^{d/2} \epsilon^{-d/2-1} (d - 2)^{-1} \right\} \\
&= 4 \left( \sqrt{\frac{32}{n}} 2^{(3d-2)/2} \left( \varepsilon \gamma^{n-1} \mathbf{Lip}_\alpha \delta_\theta Y_{\mathrm{u}}^{m-1}(t) \right)^{d/2} \frac{d+2}{4(d-2)} \right)^{(d/2+2)^{-1}},
\end{aligned} \quad (68)
$$

where we assume the data dimensionality is $d > 2$. The second line is a direct consequence of Theorem 16 (von Luxburg & Bousquet, 2004), the second inequality can be derived from the fact that $\mathbb{Q}_n$ is an empirical measure, and $\mathbb{A}_m^j$ is a metric ball of radius $\mathbf{r}_m$ embedded on finite-dimensional Hilbert space following by (**H4**). By setting $d = d'$, the last result comes from the definition of radius $\mathbf{r}_m$. $\qquad \square$

**Proposition 4.2.** *(Sampling Complexity) Let $\nu_t^N, \hat{\mu}_t$ probability measures defined in Eq (7). Then, there exist numerical constants $\mathfrak{c}, \mathfrak{c}_7, \mathfrak{c}_8, \mathfrak{c}_9 > 0, w > 0$ and $\kappa > 0$ such that the probability of squared 2-Wasserstein distance can be controlled as follows:*

$$\mathbb{P}\left[W_2^2(\nu_t^N, \hat{\mu}_t) \geq \epsilon\right] \leq \mathfrak{A}\left(\frac{1}{\epsilon^2} e^{-N\epsilon^2/4\mathfrak{c}} + \frac{1}{N} e^{-N\epsilon}\left(1 - \frac{128 w \mathfrak{h}(\boldsymbol{\alpha})}{N}\right)^{-d/8} + \frac{1}{72^4 \epsilon \sqrt{N}}\right), \quad (69)$$

$$\mathfrak{A} = \max\left(\mathfrak{c}_9, \frac{2\mathfrak{c}_7^{3/2}}{\kappa} \exp(\mathfrak{c}_4 e^{\frac{1}{2}\mathfrak{c}_1 T})\left(e^{\kappa T} - 1\right), \mathfrak{c}_9 \exp(-4\mathfrak{c}_8)\right), \quad (70)$$

*where $u \in \mathbb{O}$, $t \in \mathbb{T}$ is arbitrary and $\mathfrak{h}(\boldsymbol{\alpha}) = \|W_{\boldsymbol{\alpha}}\|_{\mathfrak{g}}$ is a cut-norm[a] of the proposed graphons (i.e., exponential, cosinusoidal).*

---
[a]Eq. 58 clarifies the explicit upper-bound of the cut-norm for the proposed graphons.

**Remark.** The approach used in the proof to establish the concentration bound is largely inspired by the series of works on the measure concentration (Bolley et al., 2007; Budhiraja & Fan, 2017; Bayraktar & Wu, 2022; Bayraktar et al., 2023; Bayraktar & Wu, 2023), with slight modifications tailored to the structure of the proposed mean-field system. We intentionally omit some parts of the proofs in this work that have already been covered in the reference.

*Proof.* We divide the proof into separate steps.

**1. Estimation of Concentration Inequality.** For the controlled mean-field system via neural agents $\boldsymbol{\alpha}$, fix the the population $\boldsymbol{\nu}^{\boldsymbol{\alpha}}$ and its related control $\mathbf{X}_u^{\nu,\boldsymbol{\alpha}} = \mathbf{X}_u$ and let $u = i/n$ for the moment. First, let us define the following probability measures:

$$\nu_t^n := \frac{1}{n}\sum_i^n \delta_{\mathbf{X}_i^n(t)}, \quad \bar{\nu}_t^n := \frac{1}{n}\sum_i^n \delta_{\mathbf{X}_{(i/n)}(t)}, \quad \hat{\mu}_t = \int \nu_u(t) p(du), \quad \bar{\mu}_t^n = \frac{1}{n}\sum_i^n \nu_{u=i/n}(t). \quad (71)$$

Then, we analyze the law of difference between the following two mean-field dynamics:

$$\mathbf{X}_u(t) = \mathbf{X}_u(0) + \int_0^t \langle \mathbb{W}_{\boldsymbol{\alpha}}[\nu_{v,s}], \boldsymbol{\psi}\rangle(\mathbf{X}_u(s)) ds + \int_0^t \boldsymbol{b}(s, \mathbf{X}_u(s), \boldsymbol{\alpha}) ds + \int_0^t \sigma_s dW_s^u,$$

$$\mathbf{X}_i^n(t) = \mathbf{X}_{(i/n)}(0) + \int_0^t \langle \mathbb{W}_{\boldsymbol{\alpha}}[\delta_{v,s}], \boldsymbol{\psi}\rangle(\mathbf{X}_i^n(s)) ds + \int_0^t \boldsymbol{b}(s, \mathbf{X}_i^n(s), \boldsymbol{\alpha}) ds + \int_0^t \sigma_s dW_s^{(i/n)}.$$

Given that fact that the expectation of Ito's differential for mean-square error can be expressed as $d_{\mathbf{I}}\|A(t)\|^2 = 2\langle A(t), \boldsymbol{m}_A\rangle dt + 2\sigma A(t) dW_t + \sigma^2 dt$ where $\mathbb{R}^+ \ni \sigma$ and $\boldsymbol{m}_A$ are compensate and martingale part of $A(t)$, we get

$$d_{\mathbf{I}}\|\mathbf{X}_{(i/n)}(t) - \mathbf{X}_i^n(t)\|_E^2 = 2\delta\mathbf{X}(t) \cdot \left(\boldsymbol{b}(s, \mathbf{X}_i^n(s), \boldsymbol{\alpha}) - \boldsymbol{b}(s, \mathbf{X}_{i/n}(s), \boldsymbol{\alpha})\right) dt$$

$$\leq \left(\frac{1}{n}\sum_{j=1}^n W_{\boldsymbol{\alpha}}\left(\frac{i}{n}, \frac{j}{n}\right)\boldsymbol{\psi}_{\boldsymbol{\alpha}}(\mathbf{X}_i^n(t), \mathbf{X}_j^n(t)) - \hat{\mathbb{E}}\left[W_{\boldsymbol{\alpha}}\left(\frac{i}{n}, v\right)\boldsymbol{\psi}_{\boldsymbol{\alpha}}(\mathbf{X}_{(i/n)}(t), x)\right]\right) \quad (72)$$

$$\cdot 2\delta\mathbf{X}(t) dt$$

where we denote $\hat{\mathbb{E}} := \mathbb{E}_{v\sim p(v), x\sim \nu_{v=j/n}(t)}$ and $p(v) := w_{\#}[\mathbf{Unif}(\mathbb{O})]$, $\delta\mathbf{X}(t) := \mathbf{X}_{(i/n)}(t) - \mathbf{X}_i^n(t)$. Then, the dissipativity assumption gives

$$d_{\mathbf{I}}\|\delta\mathbf{X}(t)\|_E^2 \leq \mathrm{I} + \mathrm{II} + \mathrm{III} + \mathrm{IV} \quad (73)$$

For simplicity let us denote $W^{i,j} := W_{\boldsymbol{\alpha}}(i/n, i/j)$, and $W^{i,v} := W_{\boldsymbol{\alpha}}(i/n, v)$. Using the dissipativity of the proposed drift function. For the second first, one can get

$$\mathrm{I} := 2\delta\mathbf{X}(t) \cdot \left(\boldsymbol{b}(s, \mathbf{X}_i^n(s), \boldsymbol{\alpha}) - \boldsymbol{b}(s, \mathbf{X}_{i/n}(s), \boldsymbol{\alpha})\right) \leq -\mathfrak{c}_1\|\delta\mathbf{X}(t)\|_E^2 \quad (74)$$

By adding and subtracting new terms, we have

$$\text{II} := \left( \frac{1}{n} \sum_{j}^{n} W^{i,j} \left[ \boldsymbol{\psi_\alpha}(\mathbf{X}_i^n, \mathbf{X}_j^n) - \boldsymbol{\psi_\alpha}(\mathbf{X}_{(i/n)} - \mathbf{X}_{(j/n)}) \right] \right) \cdot \delta\mathbf{X}(t)$$

$$\leq \frac{\mathbf{Lip_b}}{n} \sum_{j}^{n} |\delta\mathbf{X}(t)| \left( |\delta\mathbf{X}(t)| + |\mathbf{X}_j^n(t) - \mathbf{X}_{(j/n)(t)}| \right) \tag{75}$$

Similarly, the second term can be upper-bounded as follows:

$$\text{III} := \left( \frac{1}{n} \sum_{j}^{n} W^{i,j} \left[ \boldsymbol{\psi_\alpha}(\mathbf{X}_{(i/n)}, \mathbf{X}_{(j/n)}) - \mathbb{E}_{\nu_{j/n}(t)} \boldsymbol{\psi_\alpha}(\mathbf{X}_i^n, \cdot) \right] \right) \cdot \delta\mathbf{X}(t)$$

$$\leq |\delta\mathbf{X}(t)| \cdot \|W^{i,j}\|_\infty \|\mathcal{F}_{\text{III}}^i\|_E^2. \tag{76}$$

By adding and subtracting the term $W_{i,j} \mathbb{E} \boldsymbol{\psi_\alpha}(\mathbf{X}_{i/n}(t), \cdot)$, the fourth term can be improved as

$$\text{IV} := \left( \frac{1}{n} \sum_{j}^{n} \left[ W^{i,j} \int \boldsymbol{\psi_\alpha}(\mathbf{X}_{i/n}(t), \cdot) d\nu_{i/n}(t) - \int W^{i,\text{v}} \boldsymbol{\psi_\alpha}(\mathbf{X}_{i/n}(t), \cdot) d\nu_\text{v}(t) \right] \right) \cdot \delta\mathbf{X}(t)$$

$$\leq \frac{1}{n} \sum_{j}^{n} \|W^{i,j}\|_\infty \left( C_1 \mathcal{W}_2(\nu_{i/n}(t), \nu_\text{v}(t)) + n_2 d_{\mathfrak{g}}(W^{i,j}, W^{i,\text{v}}) \right)$$

$$\leq |\delta\mathbf{X}(t)| \cdot \|W^{i,j}\|_\infty \|\mathcal{F}_{\text{IV}}^i\|_E^2 \xrightarrow{n \to \infty} 0. \tag{77}$$

Note that the last inequality tends to zero for large enough $n$. Aggregating all the terms and using the fact that $g'(t) \leq ag(t) + b$ implies $g(t) \leq \int e^{-a(t-s)} b \, ds$ and $d/dt \|g(t)\|_E^2 \leq (1/2) g(t)^{-1/2} \dot{g}(t)$, where $g(t) := (1/n) \sum_i^n \|\delta\mathbf{X}(t)\|_E^2$ and $a = (2\mathbf{Lip_b} - \mathfrak{c}_1)$, $b := b(\mathcal{F}_{\text{III}}^i, \mathcal{F}_{\text{IV}}^i)$, we have

$$\mathcal{W}_2^2(\nu_t^n, \bar{\nu}_t^n) \leq \frac{1}{n} \sum_i^n \|\delta\mathbf{X}(t)\|_E^2$$

$$\leq \int_0^t e^{-(4\mathbf{Lip_b} - 2\mathfrak{c}_1)(t-s)} \left( \sup_{i',j'} \|W^{i',j'}\|_\infty^2 \frac{1}{n} \sum_i^n \left| \|\mathcal{F}_{\text{III}}^i\|_E^2 + \|\mathcal{F}_{\text{IV}}^i\|_E^2 \right|^2 \right) ds.$$

$$\leq \underbrace{\int_0^t e^{-(4\mathbf{Lip_b} - 2\mathfrak{c}_1)(t-s)} \left( \sup_{i',j'} \|W^{i',j'}\|_\infty^2 \frac{1}{n} \sum_i^n \|\mathcal{F}_{\text{III}}^i\|_E^2 + \|\mathcal{F}_{\text{IV}}^i\|_E^2 \right) ds}_{:= \text{V} + \text{VI}} \tag{78}$$

where the first inequality follows from the estimation of Wasserstein distance for empirical measures, and the last inequality can be derived by applying AM-GM inequality.

$$\mathbb{P}\left[ W_2^2(\nu_t^n, \hat{\mu}_t) \geq \epsilon \right] \leq \mathbb{P}\left[ \underbrace{W_2^2(\bar{\nu}_t^n, \hat{\mu}_t) \geq \epsilon/2}_{:= \text{VII}} \right] + \mathbb{P}[\text{V} \geq \epsilon/4] + \underbrace{\mathbb{P}[\text{VI} \geq \epsilon/4]}_{=0, \, n \gg N}, \tag{79}$$

where the last term vanishes for small enough $\epsilon$, with large $N$.

**2. Estimation of Exponential** $e^{\lambda_{\exp}\|\mathbf{X}_\mathrm{u}(t)\|_E^2}$**.** In this step, we derive the upper bound of the exponential for the square norm of mean-field predictors. We first apply the Ito's lemma to $e^{\lambda_{\exp}\|\mathbf{X}_\mathrm{u}(t)\|_E^2}$ for arbitrary scalar $\lambda_{\exp} > 0$ and observe that

$$d_{\mathbf{I}}e^{\lambda_{\exp}\|\mathbf{X}_\mathrm{u}(t)\|_E^2} = \lambda_{\exp}e^{\lambda_{\exp}\|\mathbf{X}_\mathrm{u}(t)\|_E^2}\left(2\mathbf{X}_\mathrm{u}\cdot(\boldsymbol{b}+\boldsymbol{b}_W)dt + \sigma_t(d+2\lambda_{\exp}\|\mathbf{X}_\mathrm{u}(t)\|_E^2)dt + \sigma_t dB_\mathrm{u}\right). \tag{80}$$

where gradient and Laplace of exponential can be calculated as $\nabla e^{\lambda_{\exp}\|\mathbf{X}_\mathrm{u}(t)\|_E^2} = 2\lambda_{\exp}e^{\lambda_{\exp}\|\mathbf{X}_\mathrm{u}(t)\|_E^2}$ and $\Delta e^{\lambda_{\exp}\|\mathbf{X}_\mathrm{u}(t)\|_E^2} = 2\lambda_{\exp}e^{\lambda_{\exp}\|\mathbf{X}_\mathrm{u}(t)\|_E^2}(d+2\lambda_{\exp}e^{\lambda_{\exp}\|\mathbf{X}_\mathrm{u}(t)\|_E^2})$. Taking expectation on both sides with the dissipative condition, we can show that there exist constants $\mathfrak{c}_2 = 2\lambda_{\exp}(-\mathfrak{c}_1 + \sigma_t\lambda_{\exp})$, $\mathfrak{c}_3 = \lambda_{\exp}\sigma_t d$ that directly gives following two inequalities

$$d_{\mathbf{I}}\mathbb{E}[e^{\lambda_{\exp}\|\mathbf{X}_\mathrm{u}(t)\|_E^2}] \le \mathbb{E}\left[e^{\lambda_{\exp}\|\mathbf{X}_\mathrm{u}(t)\|_E^2}\left(\mathfrak{c}_2\|\mathbf{X}_\mathrm{u}(t)\|_E^2 + \mathfrak{c}_3\right)\right]dt + \mathbb{E}\left[\int M_s dt\right], \tag{81}$$

$$\sup_{t\le T}\|\mathbf{X}_\mathrm{u}(t)\|_E^2 \le \sup_{t\le T}\|y_\mathrm{u}\|^2 + N_t + \mathfrak{c}_1\int_0^t\|\mathbf{X}_\mathrm{u}(s)\|_E^2 ds \le \mathfrak{c}_4 e^{\mathfrak{c}_1 T}. \tag{82}$$

where the second inequality is a direct consequence of Grownall's inequality, and $M_t$ and $N_t$ denote some martingale. Applying Grownall's inequality again, we have the desired result.

$$d_{\mathbf{I}}\mathbb{E}[e^{\lambda_{\exp}\|\mathbf{X}_\mathrm{u}(t)\|_E^2}] \le (\mathfrak{c}_5 + \mathfrak{c}_6\mathbb{E}[e^{\lambda_{\exp}\|\mathbf{X}_\mathrm{u}(t)\|_E^2}])dt, \tag{83}$$

$$\mathbb{E}[e^{\lambda_{\exp}\|\mathbf{X}_\mathrm{u}(t)\|_E^2}] \le (\exp(\lambda_{\exp}\|y_\mathrm{u}\|_E^2) + \mathfrak{c}_5)\exp(\mathfrak{c}_6 T) \le (\mathfrak{e}_7)^2. \tag{84}$$

where we used inequality $e^a + e^b \le \exp\left(\max(a,b) + \ln(1+\exp(-|a-b|)\right) = (\mathfrak{e}_7)^2$ such that $a = \lambda_{\exp}\|\mathbf{X}_\mathrm{u}(t)\|_E^2 + \mathfrak{c}_6 T$, $b = \ln\mathfrak{c}_5 + \mathfrak{c}_6 T$. Note that the upper-bound of the term $\exp(\lambda_{\exp}\|y_\mathrm{u}\|_E^2)$ at initial time $t = 0$ determines the exponential integrability of the right-hand side above.

**3. Estimation of Probability** $\mathbb{P}[\mathrm{V} \ge \epsilon/4]$**.** By the exponential Markov inequality with some constant $\lambda > 0$, Jensen's inequality, we obtain

$$\mathbb{P}[\mathrm{V} \ge \epsilon/4] := \mathbb{P}\left[\int_0^t e^{-(4\mathbf{Lip}_b - 2\mathfrak{c}_1)(t-s)}\left(\sup_{i',j'}\|W^{i',j'}\|_\infty^2\frac{1}{n}\sum_i^n\|\mathcal{F}_{\mathrm{III}}^i\|_E^2\right)ds > \epsilon/4\right]$$

$$\le \frac{1}{n}\sum_i^n e^{-\lambda\epsilon/4}\mathbb{E}\left[\int_0^t e^{-(4\mathbf{Lip}_b - 2\mathfrak{c}_1)(t-s)}\right. \tag{85}$$

$$\left.\cdot \exp\left(\lambda\mathfrak{h}(\boldsymbol{\alpha})\|\frac{1}{n}\sum_j^n\boldsymbol{\psi}_{\boldsymbol{\alpha}}(\mathbf{X}_{(i/n)}, \mathbf{X}_{(j/n)}) - \mathbb{E}_{\nu_{j/n}(t)}\boldsymbol{\psi}_{\boldsymbol{\alpha}}(\mathbf{X}_{(i/n)}, \cdot)\|_E^2\right)ds\right].$$

Note that $\|\boldsymbol{\psi}_{\boldsymbol{\alpha}}(\mathrm{x},\mathrm{y})\|_E \le \mathbf{Lip}_{\boldsymbol{\psi}}(\|\mathrm{x}\|_E + \|\mathrm{y}\|_E)$ have linear growth for all $\mathrm{x}, \mathrm{y} \in \mathbb{R}^d$ by the assumptions.

$$\mathbb{E}\left[\exp\left(\lambda\mathfrak{h}(\boldsymbol{\alpha})\Big\|\frac{1}{n}\sum_j^n\boldsymbol{\psi}_{\boldsymbol{\alpha}}(\mathbf{X}_{(i/n)}, \mathbf{X}_{(j/n)}) - \mathbb{E}_{\nu_{j/n}(t)}\boldsymbol{\psi}_{\boldsymbol{\alpha}}(\mathbf{X}_{(i/n)}, \cdot)\Big\|_E^2\right)\right]$$

$$\le \mathbb{E}\left[\exp\left(\frac{2\lambda\mathfrak{h}(\boldsymbol{\alpha})\mathbf{Lip}_{\boldsymbol{\psi}}}{n}\big\|\mathbf{X}_{(i/n)}\big\|_E^2 + 2\lambda\mathfrak{h}(\boldsymbol{\alpha})\Big\|\frac{1}{n}F_{\boldsymbol{\psi}}\Big\|_E^2\right)\right] \tag{86}$$

$$\le \left(2\mathbb{E}\left[\exp\left(\frac{4\lambda\mathfrak{h}(\boldsymbol{\alpha})\mathbf{Lip}_{\boldsymbol{\psi}}}{n}\big\|\mathbf{X}_{(i/n)}\big\|_E^2\right)\right]\right)^{1/2}\left(2\mathbb{E}\left[\exp\left(2\zeta\Big\|\frac{1}{n}F_{\boldsymbol{\psi}}\Big\|_E^2\right)\right]\right)^{1/2}$$

where the last inequality can be derived by applying exponential AM-GM inequality

$$\mathbb{E}\left[\exp\left(2\zeta\left\|\frac{1}{n}F_{\psi}\right\|_E^2\right)\right] = \mathbb{E}\left[\exp\left(\left\|\frac{2\sqrt{\zeta}}{n}\mathbf{Z}\right\|_E \cdot \|F_{\psi}\|_E\right)\right]$$

$$\leq \mathbb{E}\left[\exp\left(\omega\left\|\frac{2\sqrt{\zeta}}{n}\mathbf{Z}\right\|_E^2 + \frac{1}{4\omega}\|F_{\psi}\|_E^2\right)\right] \tag{87}$$

$$\leq \left(2\mathbb{E}\left[\exp\left(\frac{8\omega\zeta}{n^2}\|\mathbf{Z}\|_E^2\right)\right]\right)^{1/2}\left(2\mathbb{E}\left[\exp\left((10n)\cdot\mathbf{Lip}_{\psi}\|F_{\psi}\|_E^2\right)\right]\right)^{1/2}\exp(\mathfrak{c}_4 e^{\mathfrak{c}_1 T})$$

$$\leq 2\mathfrak{c}_7\left(1 - \frac{16\omega\zeta}{n^2}\right)^{-\frac{d}{4}}\cdot\exp(\mathfrak{c}_4 e^{\mathfrak{c}_1 T}),$$

where $\mathbf{Z} \sim \mathcal{N}(0, \mathbf{I}_d)$ is a standard Gaussian random vector. The last inequality is a direct consequence of the property of the moment generation function. The second line can be deduced from the fact that the discretized predictors $\mathbf{X}_{(i/n)}$ and $\mathbf{X}_{(j/n)}$ are i.i.d with the selection of $\omega > 0$, $\lambda_{\exp}$ and $\zeta$ satisfying the following:

$$\frac{1}{4\omega}\|F_{\psi}\|_E^2 \leq n\cdot\mathbf{Lip}_{\psi}\left(5\|\mathbf{X}_{(i/n)}\|_E^2 + \exp(\mathfrak{c}_4 e^{\mathfrak{c}_1 T})\right) \tag{88}$$

$$\lambda_{\exp} := \max\left(\frac{4\lambda\mathfrak{h}(\boldsymbol{\alpha})\mathbf{Lip}_{\psi}}{n}, (10n)\mathbf{Lip}_{\psi}\right). \tag{89}$$

$$\zeta := 2\lambda\mathfrak{h}(\boldsymbol{\alpha}) > 0 \tag{90}$$

By aggregating all the terms, we finally have

$$\mathbb{E}\left[\exp\left(\lambda\mathfrak{h}(\boldsymbol{\alpha})\left\|\frac{1}{n}\sum_j^n\boldsymbol{\psi}_{\boldsymbol{\alpha}}(\mathbf{X}_{(i/n)}, \mathbf{X}_{(j/n)}) - \mathbb{E}_{\nu_{j/n}(t)}\boldsymbol{\psi}_{\boldsymbol{\alpha}}(\mathbf{X}_{(i/n)}, \cdot)\right\|_E^2\right)\right]$$

$$\leq 2\mathfrak{c}_7^{3/2}\left(1 - \frac{16\omega\zeta}{n^2}\right)^{-\frac{d}{8}}\cdot\exp(\mathfrak{c}_4 e^{\frac{1}{2}\mathfrak{c}_1 T}) \tag{91}$$

Thus, the probability of $V$ larger than threshold $\epsilon/4$ can be written as follows:

$$\mathbb{P}[\mathrm{V} \geq \epsilon/4] \leq \frac{2}{\kappa n}e^{-n\epsilon}\mathfrak{c}_7^{3/2}\left(1 - \frac{16\omega\zeta}{n^2}\right)^{-\frac{d}{8}}\cdot\exp(\mathfrak{c}_4 e^{\frac{1}{2}\mathfrak{c}_1 T})\left(e^{\kappa T} - 1\right), \tag{92}$$

$$\kappa = -(4\mathbf{Lip}_b - 2\mathfrak{c}_1), \quad \lambda = 4n. \tag{93}$$

**4. Estimation of Probability** $\mathbb{P}[\mathrm{VII} \geq \epsilon/2]$. Now, it remains to establish the upper bound of the probability related to VII. We modify the standard estimation of concentration probabilities of empirical measures as outlined in Bolley (2010). By the triangle inequality, the probability can be decomposed as

$$\mathbb{P}\left[\mathrm{VII} \geq \frac{\epsilon}{2}\right] \leq \mathbb{P}\left[\sup_{\substack{h\Delta\leq t\leq(h+1)\Delta \\ 0\leq h\leq\bar{M}-1}}\mathcal{W}_2^2(\bar{\nu}_t^n, \bar{\nu}_{h\Delta}^n) \geq \frac{\epsilon}{6}\right] + \mathbb{P}\left[\sup_{0\leq h\leq\bar{M}-1}\mathcal{W}_2^2(\bar{\nu}_{h\Delta}^n, \bar{\mu}_{h\Delta}^n) \geq \frac{\epsilon}{6}\right] \tag{94}$$

where the temporal interval can be also decomposed as $\mathbb{T} = [0, \Delta]\cup[\Delta, 2\Delta]\cup\cdots\cup[(M-1)\Delta, T] \subseteq \bigcup_{h=0}^{M-1}[h\Delta, (h+1)\Delta]$. The first term of the right-hand side above can be bounded as

$$\mathbb{P}\left[\sup_{h\Delta\leq t\leq(h+1)\Delta}\mathcal{W}_2^2(\bar{\nu}_{t_1}^n, \bar{\nu}_{t_2}^n) \geq \frac{\epsilon}{6}\right] \leq \mathbb{P}\left[\frac{1}{n}\sup_{0\leq t_1\leq t_2\leq\mathfrak{t}}\|\mathbf{X}_{i/n}(t_1) - \mathbf{X}_{i/n}(t_2)\|_E^2 \geq \frac{\epsilon}{6}\right]$$

$$\leq \exp\left(-n\sup_{\zeta>0}\left(\epsilon\zeta - \log\mathbb{E}\exp\left(\zeta\sup_{0\leq t_1\leq t_2\leq\mathfrak{t}}\|\mathbf{X}_{i/n}(t_1) - \mathbf{X}_{i/n}(t_2)\|_E^2\right)\right)\right) \tag{95}$$

The first line is induced as any measures $\nu_{(\cdot)}^n$ are empirical, and the next line can be induced by using Chebyshev's exponential inequality and the independence of the mean-field predictor. Denoting

$\delta\mathbf{X}_{(i/n)} := \sup_{0 \le t_1 \le t_2 \le t} \|\mathbf{X}_{i/n}(t_1) - \mathbf{X}_{i/n}(t_2)\|_E^2$ for any $t_1 \le t_2 \mathbb{T}$, we can further improve the right hand side by showing

$$\mathbb{E}\exp\left(\zeta\delta\mathbf{X}_{(i/n)}\right) \le \exp(\zeta^2\mathfrak{c}_8)\exp(2\zeta\delta\mathbf{X}_{(i/n)}) \le \exp(\zeta^2\mathfrak{c}_8)\left(1 + \hat{C}\Delta\right), \qquad (96)$$

where we used the fact that $ax \le a^2b + 2ax$ for all $a, b, x \ge 0$. In order to show the upper bound of the first term in the last inequality (96), we used the result (4.6) Bolley (2010) tailored to our case under the assumption made in Section A.3 for fixed u and $\boldsymbol{\alpha}$. Combining results, we have

$$\mathbb{P}\left[\sup_{\substack{h\Delta\le t\le(h+1)\Delta \\ 0\le h\le\bar{M}-1}} \mathcal{W}_2^2(\bar{\nu}_t^n, \bar{\nu}_{h\Delta}^n) \ge \frac{\epsilon}{6}\right] \le \bar{M}\exp\left(-n\sup_{\zeta>0}\left(\epsilon\zeta - \zeta^2\mathfrak{c}_8 - \log(1+\hat{C}\Delta))\right)\right)$$

$$\le \bar{M}\exp\left(-\frac{n\epsilon^2}{4\mathfrak{c}_8} - \log(1+\hat{C}\Delta)\right) \le \frac{\mathfrak{c}_9}{\epsilon^2}\exp\left(-\frac{n\epsilon^2+1}{4\mathfrak{c}_8}\right), \quad \begin{cases} \Delta = \exp(4\mathfrak{c}_8^{-1})\hat{C}^{-1}, \\ \bar{M} \le \mathfrak{c}_9/\epsilon^2. \end{cases} \qquad (97)$$

For the second term of the right-hand side in (94), we first apply Boole's inequality of events to have

$$\mathbb{P}\left[\sup_{0\le h\le\bar{M}-1} \mathcal{W}_2^2(\bar{\nu}_{h\Delta}^n, \hat{\mu}_{h\Delta}) \ge \frac{\epsilon^2}{36}\right] \le \overbrace{\mathbb{P}\left[\sup_{0\le h\le\bar{M}-1} \mathcal{W}_2^2(\bar{\mu}_{h\Delta}^n, \hat{\mu}_{h\Delta}) \ge \frac{\epsilon^2}{72}\right]}^{\to 0, n\gg N}$$

$$+ \mathbb{P}\left[\sup_{0\le h\le\bar{M}-1} \mathcal{W}_2^2(\bar{\nu}_{h\Delta}^n, \bar{\mu}_{h\Delta}^n) \ge \frac{\epsilon^2}{72}\right] \qquad (98)$$

$$\le \frac{\bar{M}\epsilon}{(72)^4\sqrt{n}} \le \frac{\mathfrak{c}_9}{(72)^4\epsilon\sqrt{n}}.$$

The second inequality can be deduced by the result of Theorem 1.5 Bolley (2010) with $d \le d' = 4, (0,1) \ni \hat{\delta} = 2, p = 2, q = 4$. Then, there exists a constant $n_0 > 0$ such that $n \ge n_0\max\left(\epsilon^{-16}, \epsilon\right)$ for any $\epsilon > 0$ and

$$\sup_{\substack{t\in\mathbb{T} \\ i\le N}} \mathbb{P}\left[W_2^2(\delta_{\mathbf{X}_{(i/n)}(t)}), \nu_{(i/n)}(t) \ge \frac{\epsilon^2}{72}\right] \le \frac{\epsilon}{(72)^4\sqrt{n}}. \qquad (99)$$

where the quantity in (100) can be derived by proceeding similarly as in Step 2.

$$\sup_{\substack{t\in\mathbb{T} \\ i\le N}} \mathbb{E}\left[\|\mathbf{X}_{(i/n)}(t)\|_E^4\right] \le \infty \qquad (100)$$

The first term in the first inequality is direct consequence of following result:

$$\mathbb{E}\left[\|\mathbf{X}_{(i/n)}(t) - \mathbf{X}_{(i/n)}(s)\|_E^2\right] \propto |t-s|^2. \qquad (101)$$

Combining all the results for the probability bounds of V, VII for deduce the upper bound in (79),

$$\mathbb{P}\left[W_2^2(\nu_t^n, \hat{\mu}_t) \ge \epsilon\right] \le \frac{\mathfrak{c}_9}{(72)^4\epsilon\sqrt{n}} + \frac{\mathfrak{c}_9}{\epsilon^2}\exp(-4\mathfrak{c}_8)\exp\left(-\frac{n\epsilon^2}{4\mathfrak{c}_8}\right)$$

$$+ \frac{2}{\kappa n}e^{-n\epsilon}\mathfrak{c}_7^{3/2}\left(1 - \frac{128\omega\mathfrak{h}(\boldsymbol{\alpha})}{n}\right)^{-\frac{d}{8}} \cdot \exp(\mathfrak{c}_4 e^{\frac{1}{2}\mathfrak{c}_1 T})\left(e^{\kappa T} - 1\right). \quad (102)$$

By setting $\mathfrak{A}$ as follows, the proof is complete.

$$\mathfrak{A} = \max\left(\mathfrak{c}_9, \frac{2\mathfrak{c}_7^{3/2}}{\kappa}\exp(\mathfrak{c}_4 e^{\frac{1}{2}\mathfrak{c}_1 T})\left(e^{\kappa T} - 1\right), \mathfrak{c}_9\exp(-4\mathfrak{c}_8)\right). \qquad (103)$$

$\square$

## A.5   ALGORITHM

---

**Algorithm 1** Sampling Mean-field Continuous Sequence Predictors

---

**while** $t \in \mathbb{T}$ **do**               ▷ Graphon Mean-field Euler-Maruyama Sampling

    **while** $i \leq N$ **do**

        $\{y_{\mathrm{u}_i}\}_{i \leq N} \sim p(\mathrm{u}, \mathrm{y}), \Delta_t \sim p(\Delta_t), U \sim \mathbf{Unif}(\mathbb{O}), t \sim p(t)$.

$$\boldsymbol{\alpha}_i = \alpha(t, \mathbf{X}_i^n; \theta^*), W_{ij} = W_{\boldsymbol{\alpha}_i}(\lceil n\mathrm{u}_i \rceil /n, \lceil n\mathrm{v}_j \rceil /n), \boldsymbol{\psi}_{ij} = \boldsymbol{\psi}_{\boldsymbol{\alpha}_i}\left(\mathbf{X}_i^n(t), \mathbf{X}_j^n(t)\right). \quad (104)$$

$$\mathbf{X}_i^n(t + \Delta_t) = \mathbf{X}_i^n(t) + \frac{1}{n}\sum_j^n W_{ij}\boldsymbol{\psi}_{ij}\Delta_t + \boldsymbol{b}(t, \mathbf{X}_i^n, \boldsymbol{\alpha}_i)\Delta_t + \mathcal{N}(0_d, \sigma_t \Delta_t \mathbf{I}_d). \quad (105)$$

    **end while**                                ▷ Predict Subsequent Future Event

    **if** $t \in \mathbb{T} \setminus \mathbb{O}$ **then**

        $\Lambda_{t+\Delta_t} = \sum_i^K w(U, \lceil n\mathrm{u}_i \rceil /n)\mathbf{X}_i^{n, \boldsymbol{\alpha}_i}(t + \Delta_t) \approx \mathbb{E}_{\mathrm{u} \sim p(\mathrm{u})}\mathbf{X}_{\mathrm{u}}^{\boldsymbol{\alpha}}(t + \Delta_t)$

    **end if**

**end while**

---

**Graphon Mean-field Euler-Maruyama Sampling.** Algorithm 1 describes a discretization of the proposed infinite-order mean-field system. For a set of sampled temporal states, the proposed sampling method in Eq. (104) firstly projects the original graphon $W_{\boldsymbol{\alpha}}$ and interaction functions $\boldsymbol{\psi}$ onto their discrete counterparts $W_{ij}$ and $\boldsymbol{\psi}_{ij}$, which are referred to as *step graphons* Fabian et al. (2023) and *step interactions* in the literature. In the second phase, once the projections have been obtained, the Euler-Maruyama method is utilized to sample the trajectories of mean-field SDEs, effectively propagating information. In the prediction interval *i.e.*, $\mathbb{T} \setminus \mathbb{O}$, the aggregation function $w$ is utilized to integrate the sampled particles, facilitating the generation of a forecast.

**Continuity of Temporal States**. It is worth noting that every temporality integrated into the proposed framework is completely non-uniform, as minimal temporal granularity (*i.e.*, $\Delta_t$), local and global temporal states (*i.e.*, $\mathrm{u}, t$) are distributed to their corresponding probability densities defined on the continuous interval, resulting continuous representations of sequences. To operate in the described continuous setting, the neural network architecture in our framework is temporally resolution-free, which differs by exiting benchmarks, *e.g.*, Contiformer. The following list summarizes

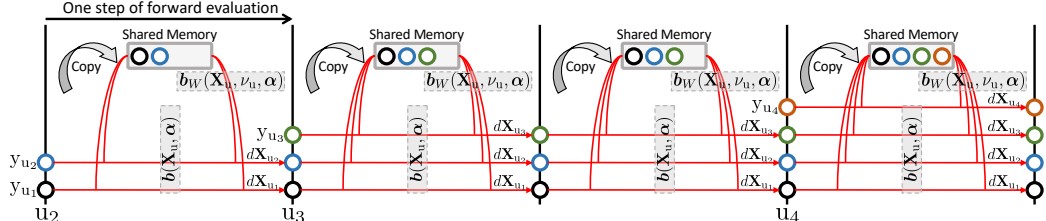

Figure 6: Parallel Computation in Sampling Mean-field Predictors

## A.6 EXPERIMENTAL DETAILS

**Experimental Setup.** We consider $\mathbb{T} := [0, T]$ as the entire temporal interval for each sequence instance and use the first $\alpha\%$ of observations, $[0, (\alpha T/100)]$, to predict the remaining $(1 - \alpha)\%$, $[(\alpha T/100), T]$. For this study, $T$ is set to 100 for the MIT Humanoid Robot dataset, 48 for MIMIC-II, 72 for the Beijing Air Quality dataset, and 1200 for Eigenworm, with $\alpha = 80$ in all cases. For every dataset we construct such input output windows on the continuous trajectories and apply per-feature $z$-score normalization using statistics computed on the training split, which are then reused for validation and test, and all models are evaluated primarily by mean squared error (MSE) on the prediction horizon. Our MFP is trained with the Adam optimizer using batch size 128 and learning rate $10^{-4}$; the neural graphon and control networks together contain on the order of one million parameters, while competing methods use the architectures and hyperparameters recommended in their original papers rather than being artificially matched by parameter count. All models, including baselines, are trained for $10\,000$ epochs without early stopping, and reported results are averaged over multiple runs with different random seeds on a single GPU. In our mean-field game interpretation, the fictitious-play index $m$ coincides with the number of Adam gradient steps on the control parameters $\alpha$, so that these $10\,000$ epochs realize $10\,000$ fictitious-play updates and provide a high-resolution approximation of the HJB component in Eq. (5) for this high-dimensional setting.

**Model Architecture.** In each forward step of $\mathbf{X}_u(t)$ from $t$ to $t + \Delta t$, a neural network takes $\mathbf{X}_u(t)$, $t$, and u as inputs and outputs $b(\cdot, \alpha)$, $W(\alpha)$, and $w$. In the first stage of the neural network, $\mathbf{X}_u(t)$ and $t$ are concatenated into a single vector, which is then projected into a hidden vector via a multilayer perceptron (MLP). This hidden vector is subsequently passed through a computation block consisting of several MLP layers with skip connections. Finally, after the computation block, the hidden vector is projected into $b(\cdot, \alpha)$, $W(\alpha)$, and $w$ using respective MLPs. In our architecture, each MLP is composed of two linear layers, with a Swish activation function positioned between them.

To process the labeling information u in the neural network, we apply adaptive normalization (Peebles & Xie, 2023). Specifically, instead of using fixed scale and shift parameters in the normalization layers of $\alpha(.; \theta)$, we regress these parameters based on u. The adaptive normalization layers are placed between MLP layers. We find that this conditioning mechanism effectively incorporates the labeling information, outperforming the approach of simply concatenating u into input vectors.

After obtaining outputs from the neural networks, we evaluate $b_W(\cdot, \alpha)$ for forward evaluation of SDEs. To derive $b_W(\cdot, \alpha)$, we compute an exponential or cosine graphon $W$ using u and v where $v < t$. Next we calculate the projection $\mathbf{Proj}(x - y) := (x - y)/\|x - y\|$ with $x = \mathbf{X}_u(t)$ and $y = \mathbf{X}_{v<t}(t)$. These values are then integrated into with $W(\alpha)$ using Eq (2) into $W_\alpha$ and $\psi_\alpha$, finally leading to $b_W(\cdot, \alpha) = \sum_{v<t} \psi_\alpha(\mathbf{X}_u(t), \mathbf{X}_v(t)) W_\alpha(u, v)$. After forward evaluation, we utilize $w$ to aggregate predictors by applying softmax. (*i.e.*, $\Lambda_t = \sum_{v<t} \texttt{Softmax}(w(u, v); \{w(t, u)\}_{u<t}) \mathbf{X}_u(t)$ where $\texttt{Softmax}(x \in S; S)$ represents the value of x after applying the softmax operation to the entire set $S$ which includes x.)

**Parallel Computation.** Since the direct application of Alg. 1 is computationally intractable for large particle count $N$, we introduce novel parallel computing to efficiently sample proposed mean-field predictors, as described in Fig **??**. At each step of forward evaluation, given all predictors $\mathbf{X}_u^\alpha$, each predictor can be processed independently using Eq (1). In other words, no predictor needs to wait for the others to complete their forward evaluation. By taking advantage of this property, at time $t$, we store all predictors with $u \le t$ in the shared memory and forward predictors one step in parallel. This

parallel implementation significantly decreases empirical computation time by reducing the number of iterations for forward evaluation from $\mathcal{O}(SN)$ to $\mathcal{O}(S)$ where $S$ is the number of steps for forward evaluation and $N$ is the number of sampled observations.

