# OpenReview forum: "Mean-Field Neural Differential Equations: A Game-Theoretic Approach to Sequence Prediction"
_ICLR.cc/2026/Conference — ICLR 2026 Conference Withdrawn Submission_

### Official Review · Reviewer_pvyM · 2025-10-28

**Soundness:** 3
**Presentation:** 3
**Contribution:** 3
**Rating:** 6
**Confidence:** 1

**Summary:**

Unable to assess the work's impact/relevance, abstaining from reviewing process

**Strengths:**

Unable to assess the work's impact/relevance, abstaining from reviewing process

**Weaknesses:**

Unable to assess the work's impact/relevance, abstaining from reviewing process

**Questions:**

Unable to assess the work's impact/relevance, abstaining from reviewing process

---

> ### Author Response · Authors · 2025-11-16
>
> We thank the reviewer for taking the time to assess our submission. Since our work may be somewhat outside standard sequence-modeling pipelines, let us briefly restate its main goal: our aim is to provide a continuous-time, mean-field–based framework that treats sequence prediction as the collective behavior of a large population of interacting predictors, and to analyze how this population behaves in the extreme regime of very fine temporal resolution. Beyond empirical performance, the central purpose of the model is to offer a principled limiting description (via mean-field games and propagation-of-chaos–type results) of what happens as the effective number of events grows, a perspective that we hope will be useful to both theory- and application-oriented readers.

---

### Official Review · Reviewer_t7CR · 2025-10-28

**Soundness:** 3
**Presentation:** 2
**Contribution:** 3
**Rating:** 6
**Confidence:** 3

**Summary:**

This paper proposes a Mean-Field Neural Differential Equation (MFP) framework that reformulates continuous-time sequence forecasting as a mean-field game, and introduces a gradient-update algorithm based on forward–backward stochastic differential equations (FBSDEs) to compute Nash equilibria. On the theoretical side, it studies the coupled partial differential equations—Hamilton–Jacobi–Bellman (HJB) and Fokker–Planck–Kolmogorov (FPK)—and provides proofs of distributional convergence. Empirically, the method shows promising performance on the evaluated benchmarks, offering partial evidence of effectiveness and indicating potential research value.

**Strengths:**

1. A nontrivial methodological contribution within the neural differential equation (NDE) literature: the paper frames continuous-time sequence modeling with a coherent design that explicitly targets irregular sampling and asynchronous events.
2. Strong theoretical development with careful mathematical derivations, yielding a principled objective and training procedure; the technical depth increases the credibility of the modeling choices.
3. On the evaluated (discrete, finitely sampled) benchmarks, the method attains competitive performance, indicating practical promise for continuous-time parameterizations (notwithstanding the limitations noted below).
4. The problem is central to continuous-time ML/NDEs and is timely for the community, with potential downstream impact on temporal point processes, latent dynamics, and long-horizon forecasting.

**Weaknesses:**

1. Template noncompliance: multiple figures (e.g., Fig. 3) place captions above the panels, and Table 2 does not follow the ICLR table style; these violate camera-ready formatting requirements and must be fixed.
2. Heavy notation with scattered definitions: the symbol usage lacks a consolidated reference, hurting readability. Provide an appendix table (symbol, meaning, shape/dimension, domain, scalar/vector) to standardize notation.
3. Reproducibility gap: no code or artifacts are released. Public implementations (training scripts/configs, environment, seeds, and optionally checkpoints) are necessary to substantiate the claims and enable adoption.
4. Evidence does not support the stated extreme-regime claim (Δt→0, events→∞): the experiments operate on discretely and finitely sampled datasets, without convergence studies vs. step size/solver tolerance, stress tests in dense-event regimes, or analyses of error accumulation in long-horizon rollouts. As written, results demonstrate performance in standard discrete settings rather than the claimed limit.
5. Text–figure inconsistency: Ablation Study I states long-term prediction on EigenWorm, while Fig. 4 is titled “Ablation studies on the MIT Humanoid Robot dataset,” creating a contradiction that undermines confidence in the reporting.
6. Missing experimental setup details and pointers to the appendix: data preprocessing/windowing/normalization, evaluation protocol and metrics, solver choice and tolerances, training hyperparameters (optimizer, schedule, batch size), random seeds, compute budget, and early-stopping criteria are not clearly specified, making it hard to assess fairness and reliability.

**Questions:**

1. The introduction states the goal as modeling in the extreme regime Δt→0 and events→∞, yet the experiments and conclusions primarily show that MFPs perform well on standard discrete forecasting. Please clarify whether the actual objective is an extreme-regime continuous-time capability or competitive performance under discrete sampling.
2. Apart from weaknesses 1, 2, and 6, the remaining listed weaknesses likewise raise concerns; please clarify each or provide supporting evidence.

---

> ### Author Response · Authors · 2025-11-16
>
> $\textbf{Q1. Template noncompliance...}$
>
> $\textbf{A}.$ Thank you for catching these template issues. We apologize for not fully adhering to the ICLR formatting guidelines. In the camera-ready version we will (i) move all figure captions, including that of Fig. 3, below the corresponding panels, and (ii) reformat Table 2 to strictly follow the official ICLR table style. We appreciate the careful reading and will correct all such formatting problems.
>
> $\textbf{Q2. Heavy notation with scattered definitions...}$
>
> $\textbf{A}.$ Thank you for this helpful suggestion. We agree that the notation is heavy and that a consolidated reference will improve readability. In the revised version, we will add a notation table in appendix that systematically lists all frequently used symbols, together with their meanings, shapes/dimensions, domains, and whether they are scalars or vectors, so that readers can easily navigate the notation.
>
> $\textbf{Q3. Reproducibility gap...}$ Thank you for pointing this out. We fully agree that public artifacts are important for reproducibility and adoption. If the paper is accepted, we will release a complete codebase at camera-ready time via a public project page, including training scripts, configuration files, environment specifications, random seeds so that our results can be independently verified and easily reused.
>
> $\textbf{Q4-1. Evidence does not support the stated extreme-regime...}$ $\textbf{Q4-2. The introduction states the goal as modeling in the extreme regime...}$
>
> $\textbf{A}.$ We appreciate this remark and agree that all empirical evaluations necessarily operate on finitely and discretely sampled datasets. Our claim about the “($\Delta t \to 0$), events ($\to \infty$)” regime is not intended as an empirical statement about running simulations at literally infinitesimal step sizes, but as a description of the **theoretical target** of the proposed framework: a continuous-time, infinite-population mean-field limit that can systematically approximate arbitrarily fine temporal resolutions.
>
> 1. **Primary theoretical objective: continuous-time, extreme-regime capability.**
>    The core modeling contribution is a mean-field, continuous-time predictor defined over a continuum of observation times ($u \in \mathbb{U} \subset [0,T]$), where the observation times themselves are random. This yields an infinite-population SDE system whose law ($\hat\mu_t$) is well defined even as the effective temporal granularity becomes arbitrarily fine and the number of events tends to infinity. The results on convergence and sample complexity for the empirical measure ($\nu_t^N$) toward ($\hat\mu_t$) are precisely formulated in this limit regime: as the number of sampled predictors (N) grows and the discretization is refined, the discrete implementation provably approaches the continuous mean-field dynamics.
>
> 2. **Practical objective: strong performance under realistic discrete sampling.**
>    At the same time, any real-world dataset is finitely and discretely sampled. Thus, our experiments are necessarily carried out in the standard setting of irregular but discrete time series. The empirical objective is therefore to show that a finite-(N), discretized MFP instantiation is not only theoretically connected to the continuous-time limit above, but also competitive or superior to existing discrete forecasters on challenging benchmarks with irregular sampling with long horizons. The results demonstrate exactly this: the same architecture that arises as a principled discretization of the continuous mean-field model also performs strongly in the discrete regime that practitioners actually face.
>
> 3. **Planned clarification in the manuscript.**
>    We agree that the current phrasing in the introduction can be read as if our *empirical* goal were to directly simulate the literal ($\Delta t \to 0$) limit. In the revised version, we will explicitly separate these two layers of the contribution:
>
>    * (i) a continuous-time, infinite-population mean-field formulation that targets the extreme regime at the theoretical level; and
>    * (ii) a finite-sample, discretized implementation that is evaluated on standard irregular discrete benchmarks, backed by convergence and sample-complexity guarantees linking it to the continuous limit.
>
>    We will rephrase the “($\Delta t \to 0$), events ($\to \infty$)” discussion to emphasize that it describes the **limiting model that our algorithm consistently approximates**, while our experiments focus on the realistic discrete setting induced by available data. In summary, the continuous-time extreme regime and the discrete experimental regime are not competing objectives but two facets of the same framework: theory characterizes the limit system, and experiments validate that its finite-sample discretization is effective under practical discrete sampling.

---

> ### Author Response · Authors · 2025-11-16
>
> $\textbf{Q5. Text–figure inconsistency...}$
>
> $\textbf{A}.$ Thank you for catching this inconsistency. This is an unintentional labeling mistake on our side: **Figure 4 actually reports the ablation results on the EigenWorm dataset**, and Ablation Study I in the text is the correct description. The underlying experiments, metrics, and conclusions are all based on EigenWorm; only the figure title was mislabeled as “MIT Humanoid Robot.” In the revised version, we will fix the title and caption of Figure 4 to explicitly state that it shows ablation studies on EigenWorm and carefully check all other figures for similar inconsistencies.
>
> $\textbf{Q6. Missing experimental setup details...}$
>
> $\textbf{A}.$ Thank you for raising this concern about missing experimental details. We agree that the setup should be described more explicitly and will clarify it in the camera-ready version by adding a concise “Experimental Setup” paragraph in Section 6 and an expanded appendix. Specifically, for each dataset we construct input–output windows on the continuous trajectories (past window for conditioning and future window for prediction), apply per-feature z-score normalization using statistics computed on the training split (and reused for validation and test), and evaluate all models primarily with mean squared error (MSE) over the prediction horizon, with MAE reported only as a secondary diagnostic. Our MFP model is trained with the Adam optimizer using batch size 128 and learning rate ($10^{-4}$), where the neural graphon and control networks together contain roughly one million parameters. Because the baselines span very different architectural families (sequence models, state-space models, neural SDEs, etc.), we keep their published architectures and hyperparameters as recommended by the original authors rather than forcing artificial parameter-count matching, and we will include a short table summarizing those settings.
>
> To ensure a uniform protocol, all models (ours and baselines) are trained for 10,000 epochs without early stopping; validation sets are used only for model selection and reporting, not to terminate training early, and all reported numbers are averaged over multiple runs with different random seeds. Finally, we will explicitly note that the fictitious-play index (m) in our mean-field game analysis corresponds one-to-one to the number of Adam gradient steps on the control parameters ($\alpha$), so that the 10,000 training epochs realize 10,000 fictitious-play updates; this makes clear that obtaining an approximate solution of the HJB component in Eq. (5) in our high-dimensional setting required a substantial iterative procedure rather than a trivial one-shot optimization.

---

> ### Author Response · Authors · 2025-11-16
> **Manuscript Update**
>
> We sincerely thank the reviewers for their thoughtful and constructive feedback. Many of the comments directly shaped substantial revisions to the manuscript, clarifying our problem formulation, sharpening the theoretical claims, and improving the organization and experimental description, and we believe the paper is significantly stronger and easier to read as a result.

---

> > ### Comment · Reviewer_t7CR · 2025-11-24
> >
> > The authors' response has addressed most of my concerns; however, I will maintain my original score.

---

### Official Review · Reviewer_c1GE · 2025-10-31

**Soundness:** 3
**Presentation:** 4
**Contribution:** 3
**Rating:** 8
**Confidence:** 3

**Summary:**

This paper introduces Mean-Field Predictors (MFPs), a novel class of neural stochastic differential equation (SDE) models, for continuous-time sequence prediction. The core idea is to model the prediction problem as a mean-field game (MFG), where a continuum of predictors (agents) interact via a neural graphon to collectively predict future events. The authors develop a gradient-based training algorithm using forward-backward SDEs (FBSDEs) and provide both theoretical convergence guarantees and empirical validation.

**Strengths:**

Originality: The paper introduces a novel formulation of sequence prediction as a mean-field game, combining neural SDEs with neural graphons in a creative way.
Quality: The methodology is technically sound with solid theoretical grounding and strong empirical results across multiple datasets.
Clarity: The paper is generally well-written, with clear definitions and helpful illustrations that support understanding.
Significance: The approach is broadly applicable to complex, irregular, and continuous-time sequence data, making it impactful for several domains.

**Weaknesses:**

The paper uses \varepsilon-Nash equilibrium as a practical solution but lacks analysis on how \varepsilon impacts prediction quality or how it behaves over time.
Key recent models like continuous-time graph or attention-based architectures are missing from the comparisons.
All datasets are low-dimensional; results on high-dimensional or real-world multivariate sequences would strengthen the claims.

**Questions:**

Could you provide a more intuitive explanation of why sequence prediction benefits from being framed as a mean-field game? How does this view improve over traditional ensembling or neural SDE approaches?
How about the impact of different ensemble-based predictors? Could we use simpler alternatives?
How about the computational cost of training interacting predictors?

---

> ### Author Response · Authors · 2025-11-16
>
> $\textbf{Q1. Could you provide a more intuitive explanation...}$
>
> $\textbf{A}.$ We thank the reviewer for asking for a more intuitive explanation. One way to picture our setting is as follows: Imagine that every past time point ($u$) is a small agent who tries to forecast the future trajectory. If we train these agents independently, each one can overfit to its own local piece of history: an agent sitting near a noisy spike might become very pessimistic, while another agent in a calm region might be overly confident. There is no mechanism that forces them to be consistent with one another, especially when we add more and more time points or when the sampling becomes very dense.
>
> In the mean-field game formulation, we **do not** treat agents in isolation. Each predictor ($X_u^\alpha(t)$) feels two influences:
>
> 1. its own local information (what happened around time ($u$)), and
> 2. the **average behavior of all other agents**, encoded by the mean-field distribution.
>
> A simple analogy is traffic on a highway: each driver decides their speed based on their own car and the “flow” of traffic around them. If everyone sped up or braked without looking at the flow, you would get shocks and crashes. If everyone takes the flow into account, traffic stays coherent even when the number of cars becomes very large. In our setting, the mean-field term plays the role of traffic flow: it tells each local predictor how the rest of the time axis “collectively believes” the future will look, and each agent adjusts its prediction accordingly. This brings two concrete intuitive benefits for sequence prediction:
>
> * **Consistency when we add more time points.**
>   As we refine the temporal grid (more events, smaller ($\Delta t$)), we are essentially adding more agents to the crowd. Because each agent looks at the same mean field, the overall belief does not change chaotically when we add new points. Instead, the population prediction stabilizes toward a limit. This is exactly what our convergence results formalize, but the intuition is: *“more agents give a finer, but not contradictory, view of the same underlying future.”*
>
> * **A natural way to handle irregular sampling.**
>   Real data are often dense in some regions of time and very sparse in others. In the mean-field picture, this just means some parts of the time axis are crowded with agents and others are almost empty. Where many agents are present, the mean field is very informative, where few are present, each agent leans more on this global signal to fill in the gaps. So the model automatically shares information across time in a way that matches how densely we have actually observed the system.
>
> Finally, the “game” aspect simply means that agents are not free to behave arbitrarily: they are all trying to **optimize a common long-term objective** defined at the population level. In practice, this means we are not just fitting predictions point-by-point, but learning a control rule that makes the entire crowd of predictors cooperate and produce a stable, coherent forecast as time becomes finer. In short, framing sequence prediction as a mean-field game helps because it turns a collection of local predictors into a **coordinated population**: each time point listens to the others through the mean field, which (i) stabilizes predictions as the temporal resolution increases and (ii) provides a simple, intuitive way to borrow strength across irregularly sampled times.

---

> ### Author Response · Authors · 2025-11-16
>
> $\textbf{Q2. How does this view improve over traditional ensembling...}$
>
> $\textbf{A}.$ We appreciate this follow-up question, because it helps clarify what is genuinely new in our perspective beyond “just using many predictors.”
>
> ---
>
> **How is this different from a traditional ensemble?**
>
> A standard ensemble (e.g., bagging, random seeds, or independently trained neural nets) typically has two features:
>
> 1. Each member is trained **largely independently** (or with very weak coupling, such as sharing a loss on the averaged prediction).
> 2. There is **no meaningful limit** as the number of members grows: adding more models can reduce variance, but the ensemble is still just a finite list of separate predictors.
>
> In our setting, the “ensemble” of predictors indexed by (u) is not just a bag of independent models:
>
> * All predictors (${X_u^\alpha(t)}_{u}$) are **driven by a single controlled SDE**, and
> * They are coupled **through the mean field** (via the neural graphon and interaction kernel), so that each agent’s update depends on the empirical distribution of all the others, not just its own data.
>
> This has two concrete consequences:
>
> * As we increase the number of predictors (more time labels, finer sampling), we are not changing the model class arbitrarily. Rather, we are approximating a **well-defined infinite-population limit**. Our convergence and sample complexity results are stated at precisely this level: the empirical measure ($\nu_t^N$) of the finite system converges to a limit law ($\hat\mu_t$) as ($N\to\infty$). Traditional ensembles usually do not admit such a structured mean-field limit.
>
> * The **control** ($\alpha$) is optimized at the population level, through the cost ($J(\nu^\alpha,\alpha)$), rather than per-member. Intuitively, we are not just asking “does each model fit its own data?” but “does the whole population of predictors behave coherently under a shared objective?” This is exactly the mean-field game aspect that ordinary ensembling does not address.
>
> So while it is helpful to think of our construction as an “infinite ensemble,” this ensemble is **tied together by the mean-field law and by a common control problem**, rather than being a collection of independent black boxes that we average at the end.
>
> ---
>
> **How does this differ from a single neural SDE?**
>
> A standard neural SDE model (say, a latent neural SDE) typically learns **one** stochastic dynamical system that maps past to future, possibly conditioned on an input representation. Our approach uses an SDE as well, but in a different role:
>
> * We model a **family** of SDE trajectories (${X_u^\alpha(t)}_{u}$) indexed by past times ($u$),
> * Each SDE is coupled to all others through the mean field ($\nu^\alpha$), and
> * The control ($\alpha$) is learned via an FBSDE/HJB-based update linked to a mean-field game.
>
> In other words, a vanilla neural SDE gives you “one stochastic flow for the sequence,” whereas the mean-field game view gives you **a population of interacting flows that must agree through their aggregate behavior** and satisfy an equilibrium condition. This is what allows us to talk meaningfully about the limit as the temporal grid is refined: the population structure and the equilibrium tie together the behavior of all these flows across different time labels.

---

> ### Author Response · Authors · 2025-11-16
>
> $\textbf{Q3. Could we use simpler alternatives?}$
>
> $\textbf{A}.$ We agree that, at the level of pure function approximation, one could certainly consider simpler alternatives: for example, a single neural SDE, or an ensemble of independently trained predictors whose outputs are averaged. These designs can work well empirically, and our own baselines include precisely such models. What is lost, however, is exactly the kind of **theoretical prediction in the large ($N$), fine resolution regime** that our paper focuses on. In a generic ensemble or single neural SDE, there is no principled way to ask: “As we add more effective time points or make the grid denser, does the global predictive system converge to a well-defined limit, or can it become unstable or collapse?” The behavior of the overall model as (N) grows is essentially opaque. One can only probe it numerically.
>
> Our mean-field game formulation is chosen precisely to make this question answerable. By viewing each time index as an agent and coupling them through a mean-field interaction, we can describe the ($N \to \infty$) regime by a **single limiting object** (the mean-field law ($\hat\mu_t$)) and then prove that the finite system ($\nu_t^N$) converges to this limit. In other words, the architecture and the analysis are aligned: the same structure that defines the model also gives us a sharp tool to study whether the population of predictors remains coherent or degenerates as we increase (N) and refine the temporal grid.
>
> To the best of our knowledge, this kind of “does the whole time series modeling system remain well behaved as the number of effective events grows?” analysis has been largely absent in prior ML work on sequence models. Our contribution is to bring **mean-field theory** in as a surgical tool to address exactly this question. Simpler alternatives can certainly be used as empirical baselines, and we do compare against them, but they would not support the type of limiting-regime guarantees and propagation-of-chaos results that our framework is designed to provide.

---

> ### Author Response · Authors · 2025-11-16
>
> $\textbf{Q4. How about the computational cost of training interacting predictors?}$
>
> $\textbf{A}.$ We appreciate the concern about computational cost. In practice, the interacting predictors are implemented as a single batched computation on GPU: all agents share the same neural graphon and interaction kernels, so their updates are evaluated in parallel rather than sequentially. As detailed in Section A.5, we use a simple parallel computation trick (vectorized graphon interactions) so that the wall-clock training time is comparable to other large continuous-time baselines. For example, on MIMIC-II our MFP has training time similar to Contiformer on the same GPU budget ($\approx$ 20GB).

---

### Note · Authors · 2026-04-22

I have read and agree with the venue's withdrawal policy on behalf of myself and my co-authors.

---

### Meta-Review · Area_Chair_9afx · 2026-01-06

**Summary:**

The paper presents a novel mean-field formulation for continuous-time sequence prediction, with solid theoretical analysis and competitive empirical results. Unfortunately, only two meaningful reviews were submitted. However, one of the two meaningful reviews rated the paper initially at 8, while the second one rated the paper at 6. The latter reviewer responded positively to the rebuttal claiming they would maintain their score. Given that both meaningful reviews were largely positive, I recommend acceptance.

**Reviewer Concerns:**

Reviewer t7CR stated: "The authors' response has addressed most of my concerns". I believe that the questions of Reviewer c1GE were adequately addressed.

**Reviewer Scores:**

I believe both reviewers would have maintained their positive rating.

---

### Decision · Program_Chairs · 2026-01-26

Accept (Poster)